# `DINO-Foresight`: Looking into the Future with DINO

**Efstathios Karypidis**[1,3]   **Ioannis Kakogeorgiou**[1]   **Spyros Gidaris**[2]   **Nikos Komodakis**[1,4,5]

[1]Archimedes, Athena Research Center, Greece    [2]valeo.ai
[3]National Technical University of Athens    [4]University of Crete    [5]IACM-Forth

## Abstract

Predicting future dynamics is crucial for applications like autonomous driving and robotics, where understanding the environment is key. Existing pixel-level methods are computationally expensive and often focus on irrelevant details. To address these challenges, we introduce `DINO-Foresight`, a novel framework that operates in the semantic feature space of pretrained Vision Foundation Models (VFMs). Our approach trains a masked feature transformer in a self-supervised manner to predict the evolution of VFM features over time. By forecasting these features, we can apply off-the-shelf, task-specific heads for various scene understanding tasks. In this framework, VFM features are treated as a latent space, to which different heads attach to perform specific tasks for future-frame analysis. Extensive experiments show the very strong performance, robustness and scalability of our framework. Project page and code at https://dino-foresight.github.io/

## 1   Introduction

Predicting future states in video sequences is a key challenge in computer vision and machine learning, with important applications in autonomous systems like self-driving cars and robotics (Finn et al., 2016; Dosovitskiy and Koltun, 2017). These systems must navigate dynamic environments safely, yet predicting future states remains difficult—especially in complex scenarios involving multi-object interactions over long time horizons.

Recent approaches focus on generating realistic RGB future frames using latent generative modeling (Rombach et al., 2022). These methods first compress RGB data into a latent space, such as continuous (Rombach et al., 2022) or discrete (Esser et al., 2021) Variational Autoencoder (VAE) representations. Then, they train generative models—like diffusion models (Zheng et al., 2024; Gao et al., 2024; Ho et al., 2022a,b; Harvey et al., 2022; Brooks et al., 2024), autoregressive models (Hu et al., 2023; Hong et al., 2023; Kondratyuk et al., 2024; Wang et al., 2024), or masked video generation (Yu et al., 2023, 2024)—to predict future states in this compressed space. While this reduces dimensionality and improves training stability (Rombach et al., 2022), VAE latents often lack semantic alignment, making them hard to interpret or use in downstream scene understanding tasks (see Table 2). Moreover, these methods must model both low-level appearance and high-level semantics, even though decision-making systems (e.g., self-driving cars) primarily need semantic scene understanding—what objects exist and where they are. Latent generative approaches may waste capacity on irrelevant low-level details, compromising temporal semantic accuracy.

In contrast, vision foundation models (VFMs) have revolutionized scene understanding with robust, transferable features (Oquab et al., 2024; Radford et al., 2021; Venkataramanan et al., 2025; Bardes et al., 2024). This raises a key question: *can VFM features serve as a semantically rich, high-dimensional latent space for precise future prediction?*

In this work, we propose precisely this idea. Instead of predicting future low-level VAE latents, we forecast the temporal evolution of VFM features directly. This shift brings several important advantages: **(a) Beyond low-level latents.** It leverages semantically meaningful representations,

39th Conference on Neural Information Processing Systems (NeurIPS 2025).

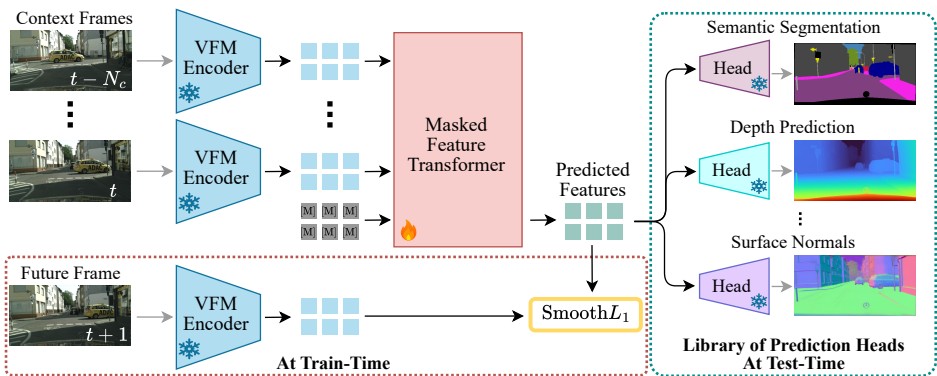

Figure 1: **Forecasting VFM Features for Future Frames.** At the core of our approach is the prediction of future VFM feature evolution. To this end, we train a masked transformer model in a self-supervised manner to forecast these features from context frames, minimizing SmoothL1 loss between predicted and actual future features. By forecasting these rich and versatile features, task-specific prediction heads—such as semantic segmentation, depth, and surface normals—can be effortlessly employed at test time, enabling modular and efficient multi-task scene understanding.

inheriting strong scene understanding. **(b) Dynamic semantics over raw frames.** It avoids full-frame synthesis, letting the model focus on meaningful dynamics, reducing complexity and sidestepping challenges like multimodal pixel distributions. **(c) Scalable multi-task support and modular integration.** Forecasted features align with downstream tasks, enabling plug-and-play integration with pretrained classifiers, segmenters or task-specific heads—without the need to retrain the core feature predictor (see Figure 1). This represents a significant departure from prior semantic feature prediction methods (Luc et al., 2017), which face significant challenges with multi-task scalability. Such approaches either require training a separate model for each task (Nabavi et al., 2018; Chiu et al., 2020; Terwilliger et al., 2019) or involve predicting features from multiple task-specific models simultaneously (Hu et al., 2020; Karypidis et al., 2025), resulting in more complex and less scalable architectures.

In this work we show that forecasting high-dimensional VFM features is not only feasible but also achieves strong performance, offering a new path toward semantically grounded future prediction.

In summary, our contributions are: **(1)** We introduce `DINO-Foresight`, a self-supervised method for semantic future prediction that builds on the key idea of forecasting VFM features—our core contribution. Unlike latent generative methods that rely on low-level VAE features, our approach delivers precise future scene understanding without modeling unnecessary appearance details, enabling a unified model for diverse scene understanding tasks. **(2)** To realize this idea, we design an efficient masked feature transformer (see Figure 1) that propagates multi-layer, high-resolution features critical for achieving strong task performance. **(3)** Experimental results demonstrate a unique advantage of our approach - our single model successfully handles multiple future-frame understanding tasks (semantic segmentation, instance segmentation, depth prediction, and surface normal prediction) where previous approaches required multiple specialized models. Furthermore, as we show in Appendix Subsection A.2 intermediate features within our masked transformer can further improve downstream task performance, highlighting its potential as a self-supervised learning strategy that enhances the already strong VFM features.

## 2 Related Work

**Video Prediction** is an extensively studied problem. Early approaches based Convolutional LSTMs (Nabavi et al., 2018; Xu et al., 2018; Wang et al., 2018; Castrejon et al., 2019; Lee et al., 2021; Wu et al., 2021; Gao et al., 2022) struggled to maintain both visual quality and temporal consistency. Subsequent developments introduced generative adversarial networks and variational autoencoders (Yan et al., 2021; Vondrick et al., 2016b; Castrejon et al., 2019; Lee et al., 2018; Babaeizadeh et al., 2018), alongside diffusion models (Ho et al., 2022a,b; Gao et al., 2024; Harvey et al., 2022), to enhance spatial-temporal coherence and improve the quality of predictions. Furthermore, transformer-

based models have also been adapted to videos, utilizing auto-regressive and masked modeling objectives to capture video dynamics (Yu et al., 2024, 2023; Gupta et al., 2023; Wang et al., 2024).

**Latent Generative Models**  Current state-of-the-art video prediction methods (Gupta et al., 2023; Yu et al., 2023, 2024; Gao et al., 2024) build upon latent generative approaches (Rombach et al., 2022). These frameworks employ an autoencoder to compress RGB frames into a latent space, then train generative models to predict future states in this compressed representation. While sharing superficial similarities with our approach—including the use of latent spaces and masked transformers in some cases (Gupta et al., 2023; Yu et al., 2023)—these methods differ fundamentally. Their latent spaces primarily encode low-level visual information, requiring reconstruction back to RGB space and forcing the model to simultaneously handle both appearance details and semantic changes.

Our key innovation lies in forecasting VFM features that natively encode high-level semantic information, enabling direct application of task-specific prediction heads. As demonstrated in Table 2, conventional VAE latent spaces cannot match this capability. Our experimental results in Table 1 further show these methods' limitations in predicting semantic scene evolution, highlighting the distinct advantages of VFM feature forecasting for scene understanding tasks. Concurrently to our work, DINO-WM (Zhou et al., 2025), also leverages DINOv2 features for world modeling with action-conditioned planning in simulated environments, while our work targets multi-task dense semantic forecasting in real world scenarios.

**Semantic Future Prediction**  An emerging approach for future-frame prediction focuses on forecasting intermediate features from an encoder rather than raw RGB values (Nabavi et al., 2018; Lin et al., 2021; Saric et al., 2020; Sun et al., 2019; Chen and Han, 2019; Jin et al., 2017; Luc et al., 2018; Šarić et al., 2019; Hu et al., 2021; Vondrick et al., 2016a; Zhong et al., 2023). This strategy models abstract encoder representations, which task-specific heads use for downstream tasks. Early methods in this paradigm include F2F (Luc et al., 2018), which regresses Mask-RCNN's feature pyramid, and F2MF (Saric et al., 2020), which improves feature prediction using flow-based warping. APANet (Hu et al., 2021) aggregates multi-level features via an auto-path mechanism for instance segmentation, while PFA (Lin et al., 2021) enhances global structures and suppresses local details for more predictable features. Recently, Futurist (Karypidis et al., 2025) introduced a multi-modal semantic forecasting approach for semantic segmentation and depth maps. While effective, these methods often rely on task- or dataset-specific encoders, limiting practicality and scalability. To address this, we use VFM encoders, which, due to large-scale pre-training, perform well across diverse tasks and generalize effectively to new scenes without retraining.

**Vision Foundation Models (VFMs)**  VFMs have transformed computer vision, achieving strong performance across a range of visual tasks. Trained on large-scale datasets, these models learn rich, transferable visual representations. Notable examples include DINOv2 (Caron et al., 2021; Oquab et al., 2024), a self-supervised model based on self-distillation; Franca (Venkataramanan et al., 2025), a fully-open VFM for scalable self-supervised representation learning; CLIP and its variants (Radford et al., 2021; Fang et al., 2023; Sun et al., 2023; Fang et al., 2024), which align visual representations with natural language; SAM (Segment Anything Model) (Kirillov et al., 2023), a foundation model for image segmentation and V-JEPA(Bardes et al., 2024), a video representation learning method via feature prediction in latent space. In this work, we explore the potential of VFM features for semantic future prediction tasks, connecting static visual understanding with dynamic prediction.

**Multi-Task Learning**  Multi-Task Learning (MTL) is a learning paradigm that enables simultaneous training of models on multiple related tasks (Maninis et al., 2019; Misra et al., 2016; Neseem et al., 2023; Vandenhende et al., 2022), promoting shared representations and improving performance across tasks. Traditional MTL frameworks often use parameter sharing (Kendall et al., 2018; Sener and Koltun, 2018; Bekoulis et al., 2018) or task interaction allowing exchange of information (Bragman et al., 2019; Chen et al., 2023a,b; Misra et al., 2016; Ruder et al., 2019). Other approaches employ a strategy that incrementally increases the model's depth during training, enabling the network to learn task-specific representations in a more resource-efficient way Aich et al. (2023); Choi and Im (2023); Guo et al. (2020); Lu et al. (2017); Zhang et al. (2022). Recently, the emergence of large-scale pretrained models has led to the introduction of adapter-based multi-task fine-tuning approaches (Liang et al., 2022; Liu et al., 2022). In our work, we leverage VFM features to provide a unified, scalable and modular framework for future prediction. Our approach enables seamless integration of multiple tasks without retraining or complex adaptations.

# 3 Methodology

Our semantic future prediction approach builds on forecasting VFM features – powerful representations that excel at scene understanding tasks and generalize well to unseen environments. We realize this through a masked transformer model trained via self-supervision to predict the temporal evolution of VFM features. This forecasted feature space serves as a latent representation that can flexibly connect to various off-the-shelf task-specific heads for future-frame analysis.

Figure 1 provides an overview of our approach, with the key components detailed in the following sections: **Section 3.1** describes how target features are generated from multi-layer VFM features. **Section 3.2** presents our formulation of VFM feature forecasting as a masked feature modeling problem and details the model architecture. **Section 3.3** covers efficient training techniques for high-resolution VFM feature prediction. Finally, **Section 3.4** introduces our modular framework for multi-task future-frame analysis and details how prediction heads are trained and integrated.

## 3.1 Hierarchical Target Feature Construction

In Figure 2, we provide an overview of how the target feature space for the feature prediction model is constructed. Below, we outline the main steps involved.

**Multi-Layer VFM Feature Extraction** Scene understanding models often benefit from processing features across multiple layers of an image encoder (Ranftl et al., 2021; Long et al., 2015; Lin et al., 2017; Cheng et al., 2022; Zhao et al., 2017; Chen et al., 2017), especially when the encoder is frozen, as in our approach. To fully leverage the pretrained VFMs, we propagate features extracted from multiple layers of the VFM. We ablate the impact of multi-layer features in Appendix Subsection A.4.

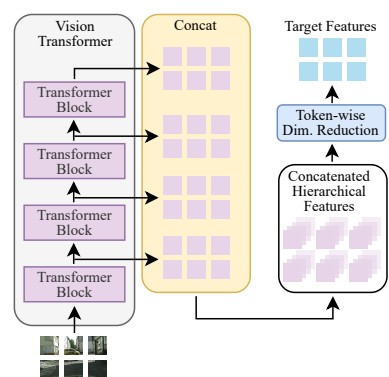

Our framework uses VFMs based on the Vision Transformer (ViT) architecture, though the approach can be extended to other architectures. Given a sequence of $N$ image frames $\mathbf{X} \in \mathbb{R}^{N \times H' \times W' \times 3}$, let $\mathbf{F}^{(l)} \in \mathbb{R}^{N \times H \times W \times D_{enc}}$ represent the features extracted from layer $l$ of the ViT model. Here, $D_{enc}$ is the feature dimension, and $H \times W$ is the spatial resolution, which are consistent across layers in ViT-based models. To form the target feature space on which the

Figure 2: **Hierarchical Target Feature Construction for the Feature Prediction Model.** Our framework constructs a feature space by extracting and concatenating multi-layer features from a frozen ViT encoder, capturing semantic information at varying abstraction levels. PCA is applied to reduce dimensionality, creating compact features.

feature prediction model operates, we concatenate the features from $L$ layers along the channel dimension, resulting in $\mathbf{F}_{\text{concat}} \in \mathbb{R}^{N \times H \times W \times L \cdot D_{enc}}$. These concatenated features capture rich semantic information from the input images at multiple levels of abstraction.

**Dimensionality Reduction** The concatenated features $\mathbf{F}_{\text{concat}}$ have high dimensionality, so we apply dimensionality reduction to simplify the feature prediction task while retaining essential information. In this work, we use Principal Component Analysis (PCA) to reduce the dimensionality, transforming $\mathbf{F}_{\text{concat}}$ into a lower-dimensional representation $\mathbf{F}_{\text{PCA}} \in \mathbb{R}^{N \times H \times W \times D}$, where $D \ll L \cdot D_{enc}$. These PCA-reduced features, $\mathbf{F}_{\text{PCA}}$, serve as the target features on which the future feature prediction operates, i.e., $\mathbf{F}_{\text{TRG}} = \mathbf{F}_{\text{PCA}}$. In Appendix Subsection A.1, we show that this PCA-based compression retains essential information and enhances downstream performance.

## 3.2 Self-Supervised Future Feature Prediction with Masked Transformers

**Masked Feature Transformer Architecture** Inspired by previous video generation models (Chang et al., 2022; Gupta et al., 2023; Karypidis et al., 2025), we implement the future feature prediction task using a self-supervised masked transformer architecture. The task involves predicting future frames in a video sequence consisting of $N$ frames, where $N_c$ are context frames and $N_p$ are future

frames to be predicted, such that $N = N_c + N_p$. Given the target features $\mathbf{F}_{\text{TRG}} \in \mathbb{R}^{N \times H \times W \times D}$ for these $N$ frames, the future-frame tokens are masked, and the transformer must predict these missing tokens by processing all the tokens from the entire sequence, i.e., all the $N \cdot H \cdot W$ tokens.

The architecture begins with a token embedding stage, where each token is projected from $D$ dimensions into the transformer's hidden dimension $D_{dec}$ through a linear layer. During training, the tokens corresponding to the future frames are replaced with a learnable $D_{\text{dec}}$-dimensional [MASK] vector. During inference, these [MASK] tokens are appended after the context frames. Each token also receives a position embedding to retain both temporal and spatial information across the sequence.

The tokens are then passed through a series of transformer layers. Standard self-attention layers in transformers have quadratic time complexity with respect to the number of tokens, making them computationally expensive for high-resolution, multi-frame sequences. To address this, we follow the approach from recent video transformers (Arnab et al., 2021; Gupta et al., 2023; Karypidis et al., 2025) and decompose the attention mechanism into temporal and spatial components. Temporal attention is applied across tokens with the same spatial position in different frames, capturing the dynamics over time. Spatial attention, on the other hand, operates within individual frames, focusing on spatial interactions. Thus, each transformer layer consists of a temporal Multi-Head Self-Attention (MSA) layer, a spatial MSA layer, and a feedforward MLP layer. After passing through the transformer layers, a linear prediction layer maps the output token embeddings from the hidden dimension $D_{\text{dec}}$ back to the feature dimension $D$, producing the predicted feature map $\tilde{\mathbf{F}}_{\text{TRG}} \in \mathbb{R}^{N \times H \times W \times D}$.

**Training Objective**  We frame the future-frame prediction as a continuous regression problem and optimize a self-supervised training objective based on the SmoothL1 loss between the predicted features $\tilde{\mathbf{F}}_{\text{TRG}}$ and the ground truth features $\mathbf{F}_{\text{TRG}}$ at the masked locations. The loss is defined as:

$$\mathcal{L}_{\text{MFM}} = \mathbb{E}_{x \in \mathcal{X}} \left[ \sum_{p \in \mathcal{P}} L \left( \mathbf{F}_{\text{TRG}}(p), \tilde{\mathbf{F}}_{\text{TRG}}(p) \right) \right], \tag{1}$$

where $\mathcal{X}$ denotes the training dataset, $\mathcal{P} = [N_c + 1 : N] \times [1 : H] \times [1 : W]$ represents the set of positions across the $H \times W$ spatial dimensions of the $N_p$ future frames, and $\mathbf{F}_{\text{TRG}}(p)$ and $\tilde{\mathbf{F}}_{\text{TRG}}(p)$ are the ground truth and predicted feature vectors at position $p$, respectively. $L(\cdot, \cdot)$ computes the SmoothL1 loss between two feature vectors:

$$L(x,y) = \sum_{d=1}^{D} \begin{cases} 0.5 \frac{(x_d - y_d)^2}{\beta}, & \text{if } |x_d - y_d| < \beta, \\ |x_d - y_d| - 0.5\beta, & \text{otherwise.} \end{cases} \tag{2}$$

In our experiments, we set $\beta = 0.1$.

### 3.3 Compute-Efficient Training Strategies for High-Resolution Feature Forecasting

Using high-resolution features is crucial for pixel-wise scene understanding tasks, such as segmentation or depth prediction, where low-resolution features struggle to capture small objects or fine spatial structures (Ranftl et al., 2021; Cheng et al., 2022; Chen et al., 2017; Touvron et al., 2019). To achieve good performance on these tasks, we aim to forecast VFM features extracted from frames with a spatial resolution of $H' \times W' = 448 \times 896$. For a ViT with a patch size of $14 \times 14$, as used in DINOv2 (Oquab et al., 2024) and EVA-CLIP (Fang et al., 2024), this results in feature maps with a resolution of $H \times W = 32 \times 64$, corresponding to 2048 tokens per frame.

However, training ViTs on such high-resolution inputs is computationally expensive in terms of both time and memory (Dosovitskiy et al., 2020). To address this challenge while maintaining efficient training, we explore the following strategies:

**Low-Resolution Training with High-Resolution Inference**  In this approach, we train on frames with a lower resolution of $224 \times 448$, resulting in features with a resolution of $16 \times 32$. During testing, we use high-resolution frames ($448 \times 896$) and adapt the position embeddings through interpolation. However, this strategy leads to suboptimal performance due to a distribution shift between the training and test data, which causes inaccurate feature forecasting.

**Sliding-Window Approach for High-Resolution Inference**  Inspired by sliding-window techniques used in segmentation tasks (Strudel et al., 2021), this strategy trains the model with cropped feature

maps. The ViT encoder extracts features from high-resolution frames ($448 \times 896$), producing high-resolution tokens (e.g., $32 \times 64$ for a patch size of $14 \times 14$). During training, we sample local crops of size $16 \times 32$, taken from the same spatial locations across frames. The model is trained on these cropped features. At inference time, the model processes the high-resolution features in a sliding-window manner using the same crop dimensions. This approach efficiently handles large inputs while avoiding the computational cost of full-resolution training.

**Two-Phase Training with Resolution Adaptation** This strategy employs a two-phase training process (Oquab et al., 2024; Touvron et al., 2019; Kolesnikov et al., 2020). First, the model is trained on low-resolution frames ($224 \times 448$) for several epochs, focusing on learning broad feature forecasting. Then, the model is fine-tuned on high-resolution ($448 \times 896$) for a small number of epochs. This adaptation phase improves the model's ability to handle high-resolution features without incurring the computational cost of training from scratch at the higher resolution. As shown in our experiments, both strategies are effective, but the two-phase approach yields better feature forecasting performance. This is likely because the masked transformer has access to a larger spatial context when propagating VFM features in future frames.

### 3.4 Modular Framework for Future-Frame Multi-Task Predictions

Our framework supports a library of interchangeable task-specific prediction heads that operate on the predicted future freames, enabling flexible multi-task future understanding. The design is modular: each head operates independently, allowing tasks to be added or removed without retraining the core feature prediction model.

We focus on four pixel-wise prediction tasks: semantic segmentation, instance segmentation, depth prediction, and surface normals prediction. However, our framework is general and can support other scene understanding tasks, such as object detection and panoptic segmentation. For semantic segmentation, depth prediction, and surface normals prediction, we use the Dense Prediction Transformer (Ranftl et al., 2021) (DPT) architecture, which is well-suited to our setup. DPT leverages multi-layer features from ViT-based encoders—consistent with the features predicted by our model—and progressively refines them into high-resolution predictions using convolutional fusion. For the instance segmentation task, we use a Mask2Former (Cheng et al., 2022) head.

Prediction heads can be trained directly on frozen VFM features and then applied "off-the-shelf" to future-frame features predicted by the model. Additionally, they can be trained to account for the PCA stage by applying PCA compression and decompression to the multi-layer features. This approach is useful for cases where prediction heads are trained on annotated 2D images, without requiring video data, and then added to the library for future-frame predictions.

## 4 Experiments

### 4.1 Experimental setup

**Data.** We assess our approach using the Cityscapes (Cordts et al., 2016) and nuScenes (Caesar et al., 2020) datasets, both offering video sequences from urban driving environments. The Cityscapes dataset includes 2,975 training sequences, 500 for validation, each with 30 frames captured at 16 fps and a resolution of $1024 \times 2048$ pixels. The 20th frame in each sequence is annotated for semantic segmentation with 19 classes. The nuScenes dataset comprises of 700 training scenes and 150 validation scenes, captured at a frame rate of 12 Hz, with each sequence extending for 20 seconds.

**Implementation details.** By default, we use DINOv2-Reg with ViT-B/14 as the default VFM visual encoder for our method. For the masked feature transformer we built upon (Besnier and Chen, 2023) implementation. We use 12 layers with a hidden dimension of $d = 1152$ and sequence length $N = 5$ (with $N_c = 4$ context frames and $N_p = 1$ future frame). For end-to-end training, we use the Adam optimizer (Kingma and Ba, 2015) with momentum parameters $\beta_1 = 0.9$, $\beta_2 = 0.99$, and a learning rate of $6.4 \times 10^{-4}$ with cosine annealing. Training is conducted on 8 A100 40Gb GPUs with an effective batch size of 64. We train DPT (Ranftl et al., 2021) heads for the semantic segmentation, depth prediction, and surface normals estimation tasks, and a Mask2Former (Cheng et al., 2022) head for the instance segmentation task. More implementation details in Appendix Subsection D.

Table 1: **Comparison with state-of-the-art on multiple forecasting tasks.** Methods that do not handle a task are marked with '-'. For approaches requiring separate prediction models per task (e.g., PFA), we show multiple entries (semantic/instance). ALL: mIoU of all classes. MO: mIoU of movable objects. VISTA$_{ft}$ is the VISTA model fine-tuned on Cityscapes. Compared approaches include: 3Dconv-F2F-RGB (Chiu et al., 2020), Dil10-S2S (Luc et al., 2017), F2F (Luc et al., 2018), ConvLSTM (Chiu et al., 2020), FeatReproj3D (Vora et al., 2018), Bayesian S2S (Bhattacharyya et al., 2019), 3Dconv-F2F-SEG (Chiu et al., 2020), DeformF2F (Šarić et al., 2019), LSTM AM S2S (Chen and Han, 2019), APANet (Hu et al., 2021), LSTM M2M (Terwilliger et al., 2019), IndRNN-Stack (Graber et al., 2021), DiffAttn-Fuse (Graber et al., 2022), F2MF (Saric et al., 2020), and PFA (Lin et al., 2021), Futurist (Karypidis et al., 2025) and VISTA (Gao et al., 2024).

| METHOD | SEMANTIC SEGMENTATION | | | | INSTANCE SEGMENTATION | | | | DEPTH | | SURFACE NORMALS | |
|---|---|---|---|---|---|---|---|---|---|---|---|---|
| | SHORT | | MID | | SHORT | | MID | | SHORT | MID | SHORT | MID |
| | ALL | MO | ALL | MO | AP50 | AP | AP50 | AP | $\delta_1$ | $\delta_1$ | 11.25° | 11.25° |
| 3Dconv-F2F-RGB | 57.0 | - | 40.8 | - | - | - | - | - | - | - | - | - |
| Dil10-S2S | 59.4 | 55.3 | 47.8 | 40.8 | - | - | - | - | - | - | - | - |
| F2F | - | 61.2 | - | 41.2 | 39.9 | 19.4 | 19.4 | 7.7 | - | - | - | - |
| ConvLSTM | 60.1 | - | - | - | - | - | - | - | - | - | - | - |
| FeatReproj3D | 61.5 | - | 45.4 | - | - | - | - | - | - | - | - | - |
| Bayesian S2S | 65.1 | - | 51.2 | - | - | - | - | - | - | - | - | - |
| 3Dconv-F2F-SEG | 65.5 | - | 50.5 | - | - | - | - | - | - | - | - | - |
| DeformF2F | 65.5 | 63.8 | 53.6 | 49.9 | - | - | - | - | - | - | - | - |
| LSTM AM S2S | 65.8 | - | 51.3 | - | - | - | - | - | - | - | - | - |
| CPConvLSTM | - | - | - | - | 44.3 | 22.1 | 25.6 | 11.2 | - | - | - | - |
| APANet | - | 64.9 | - | 51.4 | 46.1 | 23.2 | 29.2 | 12.9 | - | - | - | - |
| LSTM M2M | 67.1 | 65.1 | 51.5 | 46.3 | - | - | - | - | - | - | - | - |
| IndRNN-Stack | 67.6 | 60.8 | 58.1 | 52.1 | - | - | - | - | - | - | - | - |
| DiffAttn-Fuse | 67.9 | 61.2 | 58.1 | 51.7 | - | - | - | - | - | - | - | - |
| F2MF | 69.6 | 67.7 | 57.9 | 54.6 | - | - | - | - | - | - | - | - |
| PFA (semantic) | 71.1 | 69.2 | 60.3 | 56.7 | - | - | - | - | - | - | - | - |
| PFA (instance) | - | - | - | - | 48.7 | 24.9 | 30.5 | 14.8 | - | - | - | - |
| Futurist | **73.9** | **74.9** | **62.7** | **61.2** | - | - | - | - | **96.0** | **91.9** | - | - |
| Oracle | 77.0 | 77.4 | 77.0 | 77.4 | 66.2 | 40.4 | 66.2 | 40.4 | 89.1 | 89.1 | 95.3 | 95.3 |
| VISTA$_{ft}$ | 64.9 | 62.1 | 53.9 | 51.0 | 33.1 | 17.7 | 19.8 | 9.0 | 86.4 | 82.8 | 93.0 | 90.0 |
| DINO-Foresight (ours) | 71.8 | 71.7 | 59.8 | 57.6 | 50.5 | 26.6 | 27.3 | 12.6 | 88.6 | 85.4 | **94.4** | **91.3** |

**Evaluation Metrics.**    To evaluate our method's performance we use the following metrics: For **semantic segmentation**, we use mean Intersection over Union (mIoU) in two ways: (1) mIoU (ALL), which includes all semantic classes, and (2) MO-mIoU (MO), which considers only movable object classes like person, rider, car, truck, bus, train, motorcycle, and bicycle. For **instance segmentation**, we measure performance using average precision at a 0.50 IoU threshold (AP50) and also the mean average precision over IoU thresholds from 0.50 to 0.95. For **depth prediction**, we use mean Absolute Relative Error (AbsRel) and depth accuracy ($\delta_1$). For **surface normals**, we compute the mean angular error (m↓) and the percentage of pixels with angular errors below 11.25° (11.25°↑). Definitions of depth and surface normals metrics are provided in Appendix Subsection E.

**Evaluation Scenarios**    Following prior work (Luc et al., 2017; Nabavi et al., 2018), we evaluate our model on Cityscapes for **short-term prediction** (3 frames, 0.18s) and **mid-term prediction** (9 frames, 0.54s). On nuScenes, which has more static scenes (i.e., with less movement) than Cityscapes, we use **mid-term prediction** (9 frames, 0.75s) and **long-term prediction** (18 frames, 1.5s). Further details in Appendix Subsection F.

**Baselines**    We evaluate our method against three baselines. The first, the Oracle baseline, directly accesses the target future frame, establishing an upper performance bound. The second, Copy-Last, copies the most recent context frame to predict the target frame, providing a lower performance bound. Both baselines use DINOv2-Reg ViT-B encoder with DPT heads for semantic segmentation, depth prediction, and surface normals estimation, and a Mask2Former head for instance segmentation. The third baseline leverages VISTA (Gao et al., 2024), a state-of-the-art world model that uses video latent diffusion to generate future RGB frames from three context frames. This is a large-scale model, comprising 2.5 billion parameters and trained on 1,740 hours of driving videos, which we fine-tune on Cityscapes and nuScenes using the same frame-rate as our model. VISTA generates future RGB frames (with action-free conditioning), which are processed by a DINOv2-Reg encoder with DPT heads (identical to other baselines) for semantic segmentation, depth and surface normals prediction.

Table 2: **Comparison of VFM encoders across tasks**. For each encoder (DINOv2, EVA2-CLIP, SAM), we show performance on segmentation (ALL, MO), depth estimation ($\delta_1$ accuracy, AbsRel error), and surface normal prediction (m, percentage within 11.25°).

| ENCODER | METHOD | SEGMENTATION | | | | DEPTH | | | | SURFACE NORMALS | | | |
| | | SHORT | | MID | | SHORT | | MID | | SHORT | | MID | |
| | | ALL↑ | MO↑ | ALL↑ | MO↑ | $\delta_1$↑ | AbsR↓ | $\delta_1$↑ | AbsR↓ | m↓ | 11.25°↑ | m↓ | 11.25°↑ |
|---|---|---|---|---|---|---|---|---|---|---|---|---|---|
| DINOv2 | Oracle | 77.0 | 77.4 | 77.0 | 77.4 | 89.1 | .108 | 89.1 | .108 | 3.24 | 95.3 | 3.24 | 95.3 |
| | Copy Last | 54.7 | 52.0 | 40.4 | 32.3 | 84.1 | .154 | 77.8 | .212 | 4.41 | 89.2 | 5.39 | 84.0 |
| Oquab et al. (2024) | Prediction | **71.8** | **71.7** | **59.8** | **57.6** | **88.6** | **.114** | **85.4** | **.136** | **3.39** | **94.4** | **4.00** | **91.3** |
| EVA2-CLIP | Oracle | 71.0 | 69.5 | 71.0 | 69.5 | 85.2 | .123 | 85.2 | .123 | 3.37 | 94.5 | 3.37 | 94.5 |
| | Copy Last | 51.9 | 47.7 | 38.5 | 29.5 | 81.2 | .161 | 75.6 | .216 | 4.52 | 88.5 | 5.44 | 83.6 |
| Fang et al. (2024) | Prediction | 66.3 | 64.2 | 54.5 | 49.6 | 85.1 | .122 | 82.5 | .145 | 3.56 | 93.4 | 4.18 | 90.1 |
| SAM | Oracle | 69.8 | 63.9 | 69.8 | 63.9 | 84.8 | .143 | 84.8 | .143 | 3.01 | 96.0 | 3.01 | 96.0 |
| | Copy Last | 49.4 | 41.8 | 36.8 | 26.0 | 78.3 | .211 | 73.4 | .267 | 4.84 | 87.4 | 5.77 | 82.4 |
| Kirillov et al. (2023) | Prediction | 65.3 | 59.3 | 52.5 | 43.9 | 81.3 | .178 | 77.6 | .209 | 3.80 | 92.8 | 4.49 | 89.2 |
| VAE | Oracle | 47.3 | 34.7 | 47.4 | 35.2 | 61.5 | .251 | 61.5 | .251 | 5.3 | 86.1 | 5.3 | 86.1 |
| | Copy Last | 37.1 | 25.6 | 28.5 | 16.5 | 60.7 | .252 | 59.2 | .286 | 5.8 | 83.2 | 6.3 | 80.1 |
| Rombach et al. (2022) | Prediction | 33.4 | 17.9 | 24.7 | 9.8 | 64.1 | .281 | 61.4 | .394 | 6.5 | 80.5 | 8.0 | 73.2 |

## 4.2 VFM feature forecasting results

**Comparison with State-of-the-Art** Table 1 compares our method with state-of-the-art approaches for semantic/instance segmentation, depth estimation, and surface normal forecasting. The results highlight our key advantage: a single feature prediction model that achieves competitive or superior performance across multiple scene understanding tasks. In contrast, prior works either require separate prediction models per task (Luc et al., 2018) or handle at most two tasks simultaneously (Karypidis et al., 2025). This demonstrates the flexibility and practicality of our VFM feature forecasting approach.

**Unified Representations for Multiple Tasks: VFM Features vs. RGB Pixels** An alternative to forecasting VFM features is predicting future frames directly in RGB space, which also supports performing multiple downstream tasks through standard scene understanding models. For comparison, we fine-tune VISTA (Gao et al., 2024) to generate five future frames from three context frames (8 frames in total). We process these synthesized frames (both short-term and mid-term) with DINOv2-Reg ViT-B and DPT heads (similar to our method's setup) for fair evaluation. Despite VISTA's large-scale training and model size (2.5B parameters), it achieves lower performance on semantic segmentation, depth estimation, and surface normal prediction. Extended evaluations on nuScenes (provided Appendix Table 7 ) show similar trends for depth and surface normal estimation. Our approach is also far more computationally efficient: mid-term forecasting on Cityscapes' 500 validation scenes takes approximately 5 minutes versus VISTA's 8.3 hours (both on a single A100 GPU). This highlights our method's advantages of operating in VFM feature space – achieving accurate semantic prediction while being significantly more resource-efficient.

**Comparison of VFM Visual Encoders** In Table 2, we evaluate our method using three VFM encoders to extract features for our feature prediction model: DINOv2 with registers (Darcet et al., 2024; Oquab et al., 2024) (self-supervised), EVA2-CLIP (Fang et al., 2024) (vision-language contrastive), and SAM (Kirillov et al., 2023) (supervised instance segmentation). For each, we use the ViT-B variant. We also include `Copy-Last` and `Oracle` baselines for comparison. The results show that: **(1)** DINOv2 consistently outperforms other encoders across all tasks, achieving the best results for both short- and mid-term predictions. **(2)** This aligns with expectations, as the DINOv2-based `Oracle` also performs best in all cases. **(3)** Our model effectively predicts future-frame features for all VFMs, significantly improving over the `Copy-Last` baseline. Based on these findings, we select DINOv2 as our default VFM encoder.

**Future Prediction: VFM features vs VAE-based latents** Additionally, in Table 2, we evaluate using VAE latents (Rombach et al., 2022) (used in latent generative models) instead of VFM features, training DPT prediction heads on these latents. Results are significantly worse, as expected, since these latents lack high-level information, and even DPT oracles perform poorly. This highlights the advantage of forecasting VFM features over VAE latents.

Table 3: **Continuous vs. Discrete VFM Representations**. Comparison of continuous DINOv2 features (our approach) against 4M's DINOv2 tokenizer with discrete codes. Unlike other tables and for fair comparison with the 4M tokenized variant, we use the DINOv2 ViT-B w/o Reg model and extract features from the last layer only. Results on semantic segmentation forecasting.

| | CONTINUOUS | | | | DISCRETE (4M TOKENIZED) | | | |
| | SHORT | | MID | | SHORT | | MID | |
| METHOD | ALL↑ | MO↑ | ALL↑ | MO↑ | ALL↑ | MO↑ | ALL↑ | MO↑ |
|---|---|---|---|---|---|---|---|---|
| Oracle | 72.9 | 74.0 | 72.9 | 74.0 | 70.2 | 71.4 | 70.2 | 71.4 |
| Copy Last | 54.7 | 51.9 | 40.5 | 32.2 | 53.7 | 51.0 | 40.0 | 31.6 |
| Prediction | 68.9 | 69.3 | 57.3 | 55.0 | 61.7 | 60.9 | 53.7 | 51.0 |

Table 4: **Strategies for Training-Efficient High-Resolution Feature Forecasting**.

| RESOLUTIONS (Train→Test) | ADAPTATION APPROACH | SHORT-TERM ALL | MO | MID-TERM ALL | MO |
|---|---|---|---|---|---|
| Oracle | | | | | |
| (a) 224→224 | N/A | 68.24 | 66.41 | 68.24 | 66.41 |
| (b) 448→448 | N/A | 76.97 | 77.40 | 76.97 | 77.40 |
| Forecasting | | | | | |
| (c) 224→224 | N/A | 64.50 | 62.63 | 55.49 | 52.62 |
| (d) 224→448 | Pos. interp. | 64.34 | 64.29 | 48.31 | 44.60 |
| (e) 224→448$_{224}$ | Sliding win. | 71.26 | 71.11 | 58.75 | 56.78 |
| (f) (224&448)→448 | Two-phase | **71.81** | **71.71** | **59.78** | **57.65** |

**Discrete vs. Continuous VFM Representations**  Recent work in generative modeling has explored both discrete and continuous representations for image and video generation (Chang et al., 2022; Yu et al., 2023; Yan et al., 2021; Razavi et al., 2019; Li et al., 2024; Tschannen et al., 2024). Recent findings favor continuous representations (Li et al., 2024; Tschannen et al., 2024), showing that removing vector quantization can improve generation quality while retaining the benefits of sequence modeling. To further investigate this in the context of VFM feature forecasting, we employ 4M's pretrained DINOv2 tokenizer (Bachmann et al., 2024), which encodes DINOv2 features (without register tokens) into discrete codes from a vocabulary of size 8192. We train a quantized variant of DINO-Foresight (ViT-B14, single-layer features) using a cross-entropy loss to predict these discrete codes, which are decoded back to DINOv2 features at inference time. Results are reported in Table 3. While the discretized variant achieves comparable oracle performance to the continuous VFM feature case, our continuous VFM feature forecasting approach yields superior future semantic prediction results. These findings suggest that preserving the rich, continuous representations from VFMs—without quantization—offers clear advantages for dense semantic forecasting tasks.

**Training-Efficient Strategies for High-Resolution Feature Forecasting**  In Table 4, we compare the resolution-adaptation strategies from Section 3.3, reporting results for future semantic segmentation. *High-resolution features are essential*: comparing the low-resolution Oracle baseline (model (a)) with the high-resolution Oracle baseline (model (b)) highlights the importance of high-resolution features for strong segmentation performance. Consequently, forecasting low-resolution features (model (c)) results in significantly poorer segmentation than models predicting high-resolution features (models (e) and (f)). Adapting a model trained on low-resolution features for high-resolution inputs by simply adjusting position embeddings during inference (model (d)) leads to suboptimal results, even underperforming compared to low-resolution forecasting. The other two adaptation strategies—Sliding Window (model (e)) and two-phase training with resolution increase (model (f))—achieve considerably better results, demonstrating their effectiveness. The two-phase approach is simpler and yields the best performance, so we adopt it as our default for high-resolution feature forecasting.

**Additional Ablations and Analysis**  We conduct comprehensive ablation studies in the appendix to further validate our approach. First, in Appendix Subsection A.3, we demonstrate strong zero-shot generalization by training on Cityscapes and evaluating on nuScenes without fine-tuning. The resulting performance is only slightly worse than models trained directly on nuScenes, while surpassing

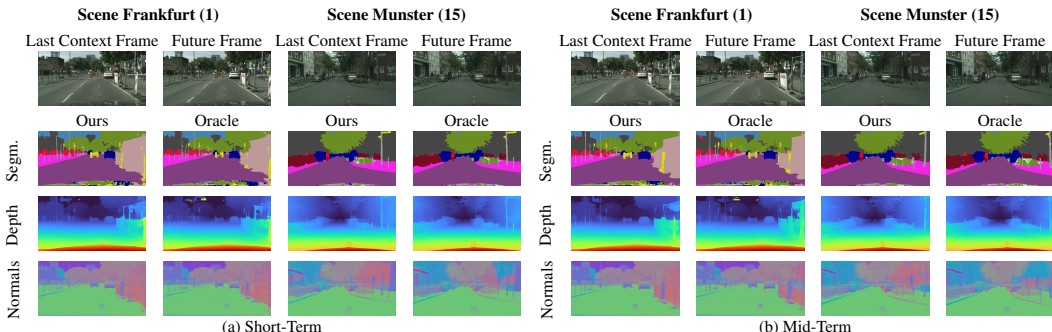

Figure 3: **Future predictions for semantic segmentation, depth, and surface normals.** Noisy segmentations at the bottom of the image (in both predicted and Oracle results) are due to unannotated regions in Cityscapes that are ignored during DPT training. This artifact affects only segmentation, not the predicted features, as evident in the clear depth and surface normal predictions.

all baselines. Second, in Appendix Subsection A.5, we investigate the impact of different masking strategies (random vs. full masking) and loss functions (L1, MSE, SmoothL1, and SmoothL1 with cosine similarity). Third, in Appendix Subsection A.6, we analyze scalability across model sizes (Small: 115M, Base: 258M, Large: 460M parameters) and data scale (Cityscapes alone vs. Cityscapes+nuScenes combined), demonstrating consistent performance improvements with increased capacity and data diversity. These results highlight the promising robustness and scalability of our VFM feature forecasting approach.

### 4.3 Qualitative results.

In Figure 3, we present qualitative results from our method applied to semantic segmentation, depth estimation, and surface normal prediction tasks, with both short-term and mid-term future predictions. Our single VFM feature prediction model produces meaningful outputs across all tasks, demonstrating the benefits of leveraging the feature space of large-scale pre-trained VFMs for future prediction.

## 5 Conclusion

In this work, we introduced `DINO-Foresight`, a self-supervised framework for semantic future prediction that shifts the paradigm from forecasting low-level latent representations to predicting high-dimensional, semantically rich VFM features. This shift offers several key advantages: it enhances scene understanding by leveraging structured semantic information, avoids the complexity of full-frame synthesis, and enables scalable, modular integration with downstream tasks.

To realize this approach, we designed a masked feature transformer that efficiently propagates high-resolution, multi-layer VFM features over time. Our experiments show that forecasting such features is not only feasible but also highly effective—demonstrating strong performance across diverse future-frame understanding tasks including semantic segmentation, instance segmentation, depth estimation, and surface normal prediction. Unlike prior methods that rely on multiple task-specific models, our single model handles all tasks seamlessly, validating the scalability and versatility of our framework.

Overall, this work lays the foundation for a new class of unified and modular future prediction systems, grounded in semantic reasoning rather than low-level reconstruction.

**Acknowledgements** This work has been partially supported by project MIS 5154714 of the National Recovery and Resilience Plan Greece 2.0 funded by the European Union under the NextGenerationEU Program. Hardware resources were granted with the support of GRNET. Also, this work was performed using HPC resources from GENCI-IDRIS (Grants 2023-A0141014182, 2023-AD011012884R2, and 2024-AD011012884R3).

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

# Appendix

## A  Additional Results

### A.1  Impact of Dimensionality Reduction

In our work, we examine a PCA-based dimensionality reduction method and find that compressing features in this way does not compromise performance on semantic segmentation and depth prediction downstream tasks (Table 5). In fact, reducing the dimensionality simplifies the modeling process and leads to improved performance. Specifically, for semantic segmentation forecasting, PCA enhances short-term predictions—particularly for moving objects—while its effect on mid-term predictions is negligible. Similarly, for depth forecasting, dimensionality reduction consistently boosts performance across all metrics for both short- and mid-term predictions.

Table 5: **Impact of Dimensionality Reduction**. Reduction is performed using PCA. Results on semantic segmentation and depth forecasting.

| DIM. REDUCTION | SEGMENTATION | | | | DEPTH | | | |
| --- | --- | --- | --- | --- | --- | --- | --- | --- |
| | SHORT | | MID | | SHORT | | MID | |
| | ALL↑ | MO↑ | ALL↑ | MO↑ | $\delta_1$↑ | AbsR↓ | $\delta_1$↑ | AbsR↓ |
| ✗ | 71.3 | 70.4 | **59.9** | **57.6** | 87.9 | .122 | 84.8 | .147 |
| ✓ | **71.8** | **71.7** | 59.8 | **57.6** | **88.6** | **.114** | **85.4** | **.136** |

### A.2  Emerging Visual Representations in the Future-Frame Masked Feature Transformer

Self-supervised representation learning has achieved remarkable progress, with numerous studies focusing on extracting robust visual features from unlabeled images and videos (Bardes et al., 2024; Tong et al., 2022; Girdhar et al., 2023; Ryali et al., 2023; Wang et al., 2023; Chen et al., 2020; Caron et al., 2021; He et al., 2022; Grill et al., 2020; Wei et al., 2022; Kakogeorgiou et al., 2024, 2022; Gidaris et al., 2024; Venkataramanan et al., 2025; Sirko-Galouchenko et al., 2025). Inspired by these advancements, we investigate the potential of our future-frame masked feature transformer as a self-supervised method for enhancing VFM visual features. Specifically, we train DPT heads for semantic segmentation and depth prediction, using not only the features predicted by the masked feature transformer but also additional features extracted from intermediate transformer layers. We examine features from the 6th, 9th, 10th, 11th, and 12th (last) layers of the transformer to assess whether these intermediate representations can further improve the strong VFM features predicted by our masked transformer.

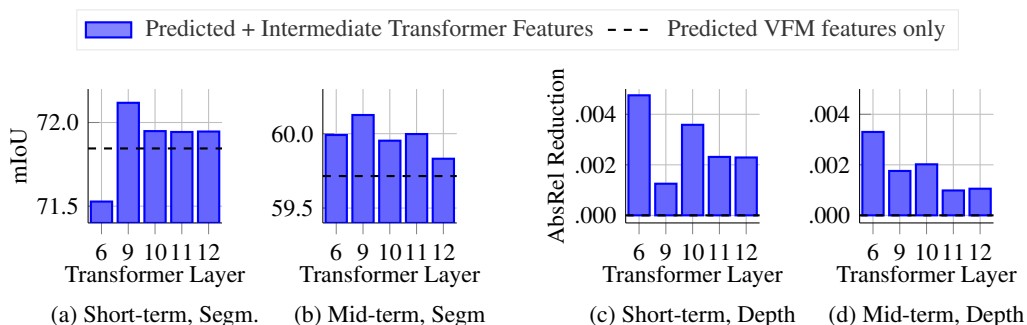

(a) Short-term, Segm.   (b) Mid-term, Segm   (c) Short-term, Depth   (d) Mid-term, Depth

Figure 4: **Impact of Intermediate Transformer Features on Future Segmentation and Depth Prediction.** Results are shown for semantic segmentation and depth prediction heads using two feature sets: only the VFM features predicted by the masked feature transformer (dashed line) and combined features from both predicted and intermediate transformer layers (blue bars). We evaluate DPT heads trained on features from the 6th, 9th, 10th, 11th, and 12th layers. For segmentation (barplots (a) and (b)), we report mIoU across all classes. For depth (barplots (c) and (d)), we show the reduction in AbsRel metric (higher is better) when adding intermediate layer features.

Results, shown in Figure 4, indicate that incorporating intermediate transformer features from the masked transformer enhances segmentation performance, with one exception (6th-layer features for short-term segmentation). Notably, the best segmentation results are achieved using features from the 9th layer. Although the improvements are modest, this aligns with expectations given the strength of the predicted VFM features alone. Similar results are observed for future depth prediction, where intermediate features also led to performance gains, with the best depth results achieved using 6th-layer features (see Figure 4). While exploring self-supervised learning was not the primary aim of our work, we find these results intriguing, as they suggest that future prediction methods hold promise as self-supervised visual representation learners. We hope this work can spark further research into this direction.

### A.3 Additional Comparisons with VISTA

In Table 6, we present a comprehensive evaluation comparing our method against VISTA (Gao et al., 2024) for semantic segmentation, instance segmentation, depth and surface normal estimation on the Cityscapes dataset. We extend this evaluation to the nuScenes (Caesar et al., 2020) dataset in Table 7 for depth and surface normals. Notably, DINO-Foresight demonstrates strong zero-shot generalization capabilities. When trained only on Cityscapes and directly evaluated on nuScenes without fine-tuning, performance degradation is minimal, while still significantly outperforming all baselines. The results show that DINO-Foresight consistently surpasses the performance of VISTA on both datasets.

Table 6: **Comparison across tasks on Cityscapes**. We use DINOv2 encoder and show performance on segmentation (ALL, MO), instance segmentation (AP50, AP), depth estimation ($\delta_1$ accuracy, AbsRel error), and surface normal prediction (m, percentage within $11.25°$). VISTA$_{ft}$ is the VISTA model fine-tuned on Cityscapes.

| METHOD | SEMANTIC SEGM. | | | | INSTANCE SEGM | | | | DEPTH | | | | SURFACE NORMALS | | | |
|---|---|---|---|---|---|---|---|---|---|---|---|---|---|---|---|---|
| | SHORT | | MID | | SHORT | | MID | | SHORT | | MID | | SHORT | | MID | |
| | ALL↑ | MO↑ | ALL↑ | MO↑ | AP50↑ | AP↑ | AP50↑ | AP↑ | $\delta_1$↑ | AbsR↓ | $\delta_1$↑ | AbsR↓ | m↓ | 11.25°↑ | m↓ | 11.25°↑ |
| Oracle | 77.0 | 77.4 | 77.0 | 77.4 | 66.2 | 40.4 | 66.2 | 40.4 | 89.1 | .108 | 89.1 | .108 | 3.24 | 95.3 | 3.24 | 95.3 |
| Copy Last | 54.7 | 52.0 | 40.4 | 32.3 | 24.7 | 10.4 | 9.5 | 2.8 | 84.1 | .154 | 77.8 | .212 | 4.41 | 89.2 | 5.39 | 84.0 |
| VISTA$_{ft}$ | 64.9 | 62.1 | 53.9 | 51.0 | 33.1 | 17.7 | 19.8 | 9.0 | 86.4 | .124 | 82.8 | .153 | 3.75 | 93.0 | 4.30 | 90.0 |
| DINO-Foresight | **71.8** | **71.7** | **59.8** | **57.6** | **50.5** | **26.6** | **27.3** | **12.6** | **88.6** | **.114** | **85.4** | **.136** | **3.39** | **94.4** | **4.00** | **91.3** |

Table 7: **Comparison across tasks at nuScenes**. We use DINOv2 encoder and show performance on depth estimation ($\delta_1$ accuracy, AbsRel error) and surface normal prediction (m, percentage within $11.25°$). VISTA$_{ft}$ is the VISTA model fine-tuned on nuScenes. The last row evaluates zero-shot generalization: DINO-Foresight trained on Cityscapes and directly evaluated on nuScenes.

| METHOD | DEPTH | | | | SURFACE NORMALS | | | |
|---|---|---|---|---|---|---|---|---|
| | MID | | LONG | | MID | | LONG | |
| | $\delta_1$↑ | AbsR↓ | $\delta_1$↑ | AbsR↓ | m↓ | 11.25°↑ | m↓ | 11.25°↑ |
| Oracle | 82.6 | .206 | 82.6 | .206 | 3.09 | 97.1 | 3.09 | 97.1 |
| Copy Last | 73.4 | .353 | 68.4 | .468 | 4.66 | 88.6 | 5.31 | 85.3 |
| VISTA$_{ft}$ | 74.6 | .337 | 70.8 | .421 | 4.46 | 90.8 | 4.96 | 88.2 |
| DINO-Foresight | **80.7** | **.218** | **76.3** | **.299** | **3.59** | **93.9** | **4.22** | **90.6** |
| DINO-Foresight (zero-shot) | 78.4 | .269 | 72.1 | .377 | 4.03 | 92.3 | 4.77 | 88.4 |

### A.4 Impact of Multi-Layer Features

Scene understanding models benefit from utilizing features from multiple layers of a frozen image encoder. To fully exploit the pretrained DINO features, we integrate representations from several layers into the DPT head. Table 8 presents an ablation comparing multi-layer DINO features to using only the final layer (layer 12) for semantic segmentation on Cityscapes. The results demonstrate that aggregating features from layers 3, 6, 9, and 12 enhances performance, with ALL (all semantic classes) and MO (movable objects classes) scores rising from 72.1/73.4 to 77.0/77.4.

Table 8: **DINO+DPT Features Ablation**. Ablation of multi-layer DINO features as input to the DPT head versus using only the last layer features, evaluated on semantic segmentation on Cityscapes.

| LAYERS | SEGMENTATION | |
|---|---|---|
| | ALL | MO |
| 12 | 72.1 | 73.4 |
| 3,6,9,12 | 77.0 | 77.4 |

## A.5 Masking strategy and loss function ablations

We evaluate the impact of masking strategies and loss functions on forecasting performance. As shown in Table 9, full masking (masking all future features) consistently outperforms random masking across all metrics, for semantic segmentation and depth forecasting. This demonstrates that masking all future features forces the model to learn more robust temporal dynamics. Regarding loss functions (Table 10), we find that our framework is robust to loss function choice, with L1, MSE, SmoothL1, and SmoothL1+Cosine achieving comparable performance.

Table 9: **Impact of Masking Strategies on Cityscapes**. We compare random masking versus full masking for semantic segmentation, depth estimation, and surface normal prediction. Full masking demonstrates consistently superior performance across all metrics.

| | SEMANTIC SEGMENTATION | | | | DEPTH | | | | SURFACE NORMALS | | | |
|---|---|---|---|---|---|---|---|---|---|---|---|---|
| MASKING STRATEGY | SHORT | | MID | | SHORT | | MID | | SHORT | | MID | |
| | ALL↑ | MO↑ | ALL↑ | MO↑ | $\delta_1$↑ | AbsR↓ | $\delta_1$↑ | AbsR↓ | m↓ | 11.25°↑ | m↓ | 11.25°↑ |
| Random Masking | 70.5 | 70.2 | 58.0 | 55.7 | 88.2 | .121 | 84.7 | .148 | 3.48 | 94.0 | 4.11 | 90.8 |
| Full Masking | **71.8** | **71.7** | **59.8** | **57.6** | **88.6** | **.114** | **85.4** | **.136** | **3.39** | **94.4** | **4.00** | **91.3** |

Table 10: **Loss Function Comparison on Cityscapes**. We evaluate different loss functions (L1, MSE, SmoothL1, SmoothL1+Cosine) for semantic segmentation, depth estimation, and surface normal prediction. Results demonstrate that our framework is robust to loss function choice, with all variants achieving comparable performance.

| | SEMANTIC SEGMENTATION | | | | DEPTH | | | | SURFACE NORMALS | | | |
|---|---|---|---|---|---|---|---|---|---|---|---|---|
| LOSS FUNCTION | SHORT | | MID | | SHORT | | MID | | SHORT | | MID | |
| | ALL↑ | MO↑ | ALL↑ | MO↑ | $\delta_1$↑ | AbsR↓ | $\delta_1$↑ | AbsR↓ | m↓ | 11.25°↑ | m↓ | 11.25°↑ |
| L1 | 71.7 | 71.7 | 59.7 | 57.6 | 88.6 | .118 | **85.6** | .138 | 3.41 | 94.3 | 4.00 | 91.3 |
| MSE | 71.7 | **71.9** | **60.0** | **57.8** | 88.6 | .117 | 85.4 | .138 | 3.40 | **94.4** | **3.99** | **91.4** |
| SmoothL1+Cos | 71.7 | 71.4 | 59.8 | 57.5 | **88.7** | .116 | 85.5 | .137 | 3.40 | **94.4** | **3.98** | **91.4** |
| SmoothL1 | **71.8** | 71.7 | 59.8 | 57.6 | 88.6 | **.114** | 85.4 | **.136** | **3.39** | **94.4** | 4.00 | 91.3 |

## A.6 Model Size and Data Scale Scalability

We investigate how performance scales with model size and training data diversity. As shown in Table 11, we evaluate three model variants—Small (115M), Base (258M), and Large (460M) parameters—by modifying the hidden dimension and number of attention heads while keeping dataset size and training duration fixed. Results demonstrate consistent performance improvements with increased model capacity, particularly for mid-term predictions, indicating that larger models better capture complex temporal dynamics in VFM features. Regarding data scale (Table 12), we combine Cityscapes and nuScenes datasets with equal-probability sampling during training and evaluate on Cityscapes. The model trained on combined datasets achieves consistent improvements across all tasks compared to training on Cityscapes alone, with gains particularly pronounced for mid-term semantic segmentation. These findings demonstrate that our framework effectively scales with both increased model capacity and diverse training data, motivating further exploration with larger models and datasets to fully realize the potential of forecasting VFM features for multi-task scene understanding.

Table 11: **Model Size Scalability on Cityscapes**. We evaluate three model sizes—Small (115M), Base (258M), and Large (460M) parameters—across semantic segmentation, depth estimation, and surface normal prediction. Results demonstrate consistent performance improvements with increased model capacity.

| Model Variant | Hidden Dim | Att Heads | Semantic Segm. | | | | Depth | | | | Surface Normals | | | |
| | | | Short | | Mid | | Short | | Mid | | Short | | Mid | |
| | | | ALL↑ | MO↑ | ALL↑ | MO↑ | $\delta_1$↑ | AbsR↓ | $\delta_1$↑ | AbsR↓ | m↓ | 11.25°↑ | m↓ | 11.25°↑ |
|---|---|---|---|---|---|---|---|---|---|---|---|---|---|---|
| Small (115M) | 768 | 6 | 71.1 | 70.8 | 59.3 | 57.3 | 87.7 | .125 | 84.8 | .142 | 3.58 | 93.7 | 4.13 | 90.8 |
| Base (258M) | 1152 | 8 | 71.8 | **71.7** | 59.8 | 57.6 | **88.6** | **.114** | 85.4 | **.136** | **3.39** | **94.4** | **4.00** | 91.3 |
| Large (460M) | 1536 | 12 | **71.9** | **71.7** | **60.2** | **58.3** | **88.6** | .116 | **85.5** | .137 | 3.40 | **94.4** | **4.00** | **91.5** |

Table 12: **Data Scale Scalability**. We compare training on Cityscapes alone versus training on combined Cityscapes+nuScenes data. Results demonstrate that increasing training data diversity leads to improved performance across all tasks and temporal horizons.

| Training Data | Semantic Segmentation | | | | Depth | | | | Surface Normals | | | |
| | Short | | Mid | | Short | | Mid | | Short | | Mid | |
| | ALL↑ | MO↑ | ALL↑ | MO↑ | $\delta_1$↑ | AbsR↓ | $\delta_1$↑ | AbsR↓ | m↓ | 11.25°↑ | m↓ | 11.25°↑ |
|---|---|---|---|---|---|---|---|---|---|---|---|---|
| Cityscapes | 71.8 | 71.7 | 59.8 | 57.6 | **88.6** | **.114** | 85.4 | **.136** | **3.39** | **94.4** | 4.00 | 91.3 |
| Cityscapes+nuScenes | **72.3** | **72.2** | **61.0** | **59.4** | **88.6** | .117 | **85.7** | **.136** | 3.41 | **94.4** | **3.95** | **91.6** |

## A.7 More Visualizations

In Figure 5 and 6, we present additional qualitative results illustrating the prediction of semantic segmentation, depth maps, and surface normals. Specifically, we compare our method, `DINO-Foresight`, against the `Oracle`, which involves using future RGB frames as inputs for different prediction heads, as well as `VISTA` Gao et al. (2024). As illustrated in Figure 5, `DINO-Foresight` maintains superior integrity of motion dynamics and geometric consistency across frames, resulting in more accurate predictions

In Figure 7 and 8, we offer additional qualitative results derived from utilizing `DINO-Foresight` for the prediction of semantic segmentation and depth maps and surface normals over extended time intervals. These outcomes are achieved through the use of autoregressive rollouts. Beginning with a series of four context frames ($X_{t-9}$ to $X_t$), the model is capable of predicting up to 48 subsequent frames, equivalent to 2.88 seconds, with predictions occurring at an interval of every third frame. Our model consistently delivers accurate predictions over the entire forecasted duration, effectively capturing motion dynamics and maintaining consistency across different modalities. This performance underscores its robustness and versatility, which are related to its capability of predicting the features of a foundation model. As a final remark, it is important to note that the noisy segmentation predictions observed at the bottom of the images, present in both the predicted and `Oracle` results, are attributed to unannotated regions in the Cityscapes dataset that are disregarded during DPT training. This artifact impacts only the segmentation outcomes of DPT head and does not affect the predicted future features, as evidenced by the clear and accurate depth and surface normal predictions.

## B Limitations and Future Work

In this work, we introduced `DINO-Foresight`, a simple yet effective method for semantic future prediction based on forecasting VFM features. Our approach delivers strong results while opening several exciting directions for future research.

First, our method uses a straightforward masked transformer with SmoothL1 loss. While forecasting VFM features avoids the challenges of modeling complex pixel distributions, our current implementation is deterministic. However, our framework can easily be extended to capture uncertainty—for example, by adding a diffusion loss (as in MAR (Li et al., 2024)) or modeling tokens with a Gaussian mixture model (as in GIVT (Tschannen et al., 2024)). These extensions would better handle future ambiguity while maintaining the simplicity of our approach.

Although we explored strategies to reduce training compute for high-resolution feature prediction, inference-time compute demands remain unchanged. Future work could address this by adopting

hierarchical transformer architectures (Ryali et al., 2023), which would not only improve efficiency but also enable the model to handle even higher feature resolutions.

Another promising direction is scaling `DINO-Foresight` to larger datasets and models. Our experiments on model size (115M to 460M parameters) and data diversity (Cityscapes+nuScenes) demonstrate consistent performance improvements, particularly for mid-term predictions, suggesting that further scaling to even larger models and more diverse datasets could yield substantial gains in forecasting performance.. Furthermore, the flexibility of our framework allows seamless integration of newer VFM encoders, such as RADIOv2.5 (Heinrich et al., 2025), which combines multiple VFMs into a single, more powerful model, enhancing its multi-task future scene understanding capabilities.

Overall, these research directions highlight the flexibility and growth potential of our approach, paving the way for further advancements in semantic future prediction.

## C  Broader Impact

Our work enables efficient and scalable semantic future prediction by forecasting semantically rich VFM features. This allows flexible integration with different scene understanding tasks without retraining, making it useful for applications like autonomous driving and robotics. While we do not foresee risks in our approach, we must remain mindful that the pretrained Vision Foundation Models we build upon may carry biases, which could potentially influence our semantic future predictions.

## D  Implementation Details

We provide implementation details for the heads trained on different downstream tasks. The DPT head is used for semantic segmentation, depth estimation, and surface normal estimation. We adopt the DPT Ranftl et al. (2021) implementation from Depth Anything Yang et al. (2024a,b), setting the feature dimensionality to 256 and configuring `dptoutchannels = [128, 256, 512, 512]`. For all tasks, models are trained for 100 epochs with a batch size of 128 ($16 \times 8$ GPUs). The learning rate is set to 0.0016, using the AdamW optimizer with linear warmup for the first 10 epochs, and weight decay is 0.0001. For semantic segmentation, we use a polynomial scheduler and cross-entropy loss with 19 classes. For depth estimation, we use a cosine annealing scheduler and cross-entropy loss, with 256 classes. For surface normal estimation, we employ a polynomial scheduler and a loss function combining cosine similarity and $L_2$ loss with weighted averaging, using 3 classes.

For the Mask2Former head used in instance segmentation, we implement our approach using the official Mask2Former Cheng et al. (2022) and Detectron2 Wu et al. (2019) codebases. The main difference compared to the official Mask2Former configuration for Cityscapes instance segmentation is the input feature maps. In our approach, the four multi-scale feature maps expected by Mask2Former are derived from the PCA features. These PCA features are first projected to 128, 256, 512, and 1024 dimensions and then resized so their spatial resolutions are $\times 4$, $\times 2$, $\times 1$, and $\times 0.5$ relative to the original resolution of the DINOv2 ViT-B outputs. We train using the AdamW optimizer, with a batch size of 64 ($8 \times 8$ GPUs), learning rate of 0.00032, weight decay of 0.05, and 67,500 iterations, with a polynomial scheduler.

Regarding Vista, we fine-tuned the model with 8 frames in total (3 as context and 5 future frames) for cityscapes, while for Nuscenes we used 9 frames in total (3 as context and 6 future frames) to support long-term forecasting

## E  Definitions of Evaluation Metrics

For **depth prediction**, we use two metrics: the mean Absolute Relative Error (AbsRel), defined as $\frac{1}{M} \sum_{i=1}^{M} \frac{|a_i - b_i|}{b_i}$, where $a_i$ and $b_i$ are the predicted and ground truth disparities at pixel $i$, and $M$ is the number of pixels. We also evaluate depth accuracy using $\delta_1$, the percentage of pixels where $\max \left( \frac{a_i}{b_i}, \frac{b_i}{a_i} \right) < 1.25$. For **surface normal evaluation**, we compute the mean angular error m↓ as $\frac{1}{N} \sum_{i=1}^{N} \cos^{-1} \left( \frac{\mathbf{n}_i \cdot \tilde{\mathbf{n}}_i}{\|\mathbf{n}_i\| \|\tilde{\mathbf{n}}_i\|} \right)$, where $\mathbf{n}_i$ and $\tilde{\mathbf{n}}_i$ are the predicted and ground truth normals,

respectively. Furthermore, we measure precision through the percentage of pixels with angular errors below 11.25°, calculated as $\left( \frac{1}{N} \sum_{i=1}^{N} \mathbb{I}(\theta_i < 11.25°) \right)$

# F  Details of Evaluation Scenarios

Regarding Cityscapes, the target frame for both short and mid term prediction is 20. We subsample sequences by a factor of 3 before inputting to the model. For short-term prediction, the model uses frames 8, 11, 14, and 17 as context to predict frame 20 (with context length $N_c = 4$ and $N_p = 1$). For mid-term prediction, the model uses frames 2, 5, 8, and 11 as context and predicts frame 20 auto-regressively through frames 14 and 17. We calculate segmentation metrics on the 20th frame using Cityscapes ground truth. For depth and surface normals, we rely on pseudo-annotations from DepthAnythingV2 Yang et al. (2024b) and Lotus He et al. (2025), respectively, due to the lack of true annotations in Cityscapes. Regarding nuScenes, the target frame for both mid and long term prediction is 29. Again, we subsample sequences by a factor of 3 before input to the model. For mid-term prediction, the model uses frames 11, 14, 17, and 20 as context and predicts frame 29 auto-regressively through frames 23 and 26. For long-term prediction, the model uses frames 2, 5, 8, and 11 as context and predicts frame 29 auto-regressively through frames 14,17,20,23 and 26. Again for depth and surface normals, we rely on pseudo-annotations from DepthAnythingV2 Yang et al. (2024b) and Lotus He et al. (2025), respectively, due to the lack of true annotations in nuScenes.

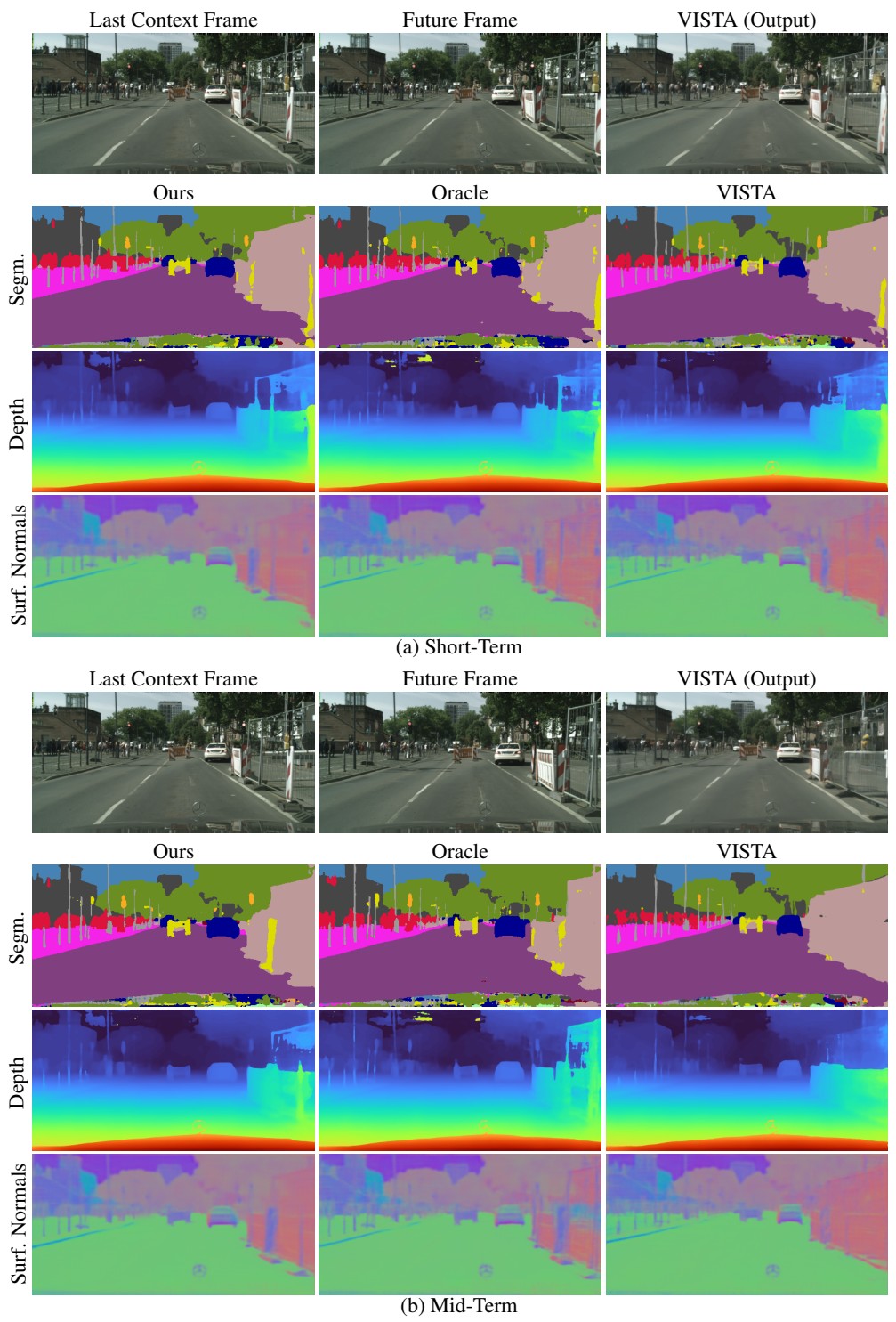

Figure 5: **Visualization of future predictions for semantic segmentation, depth, and surface normals.** The illustrated scene is Frankfurt (01 (017082-017111)).

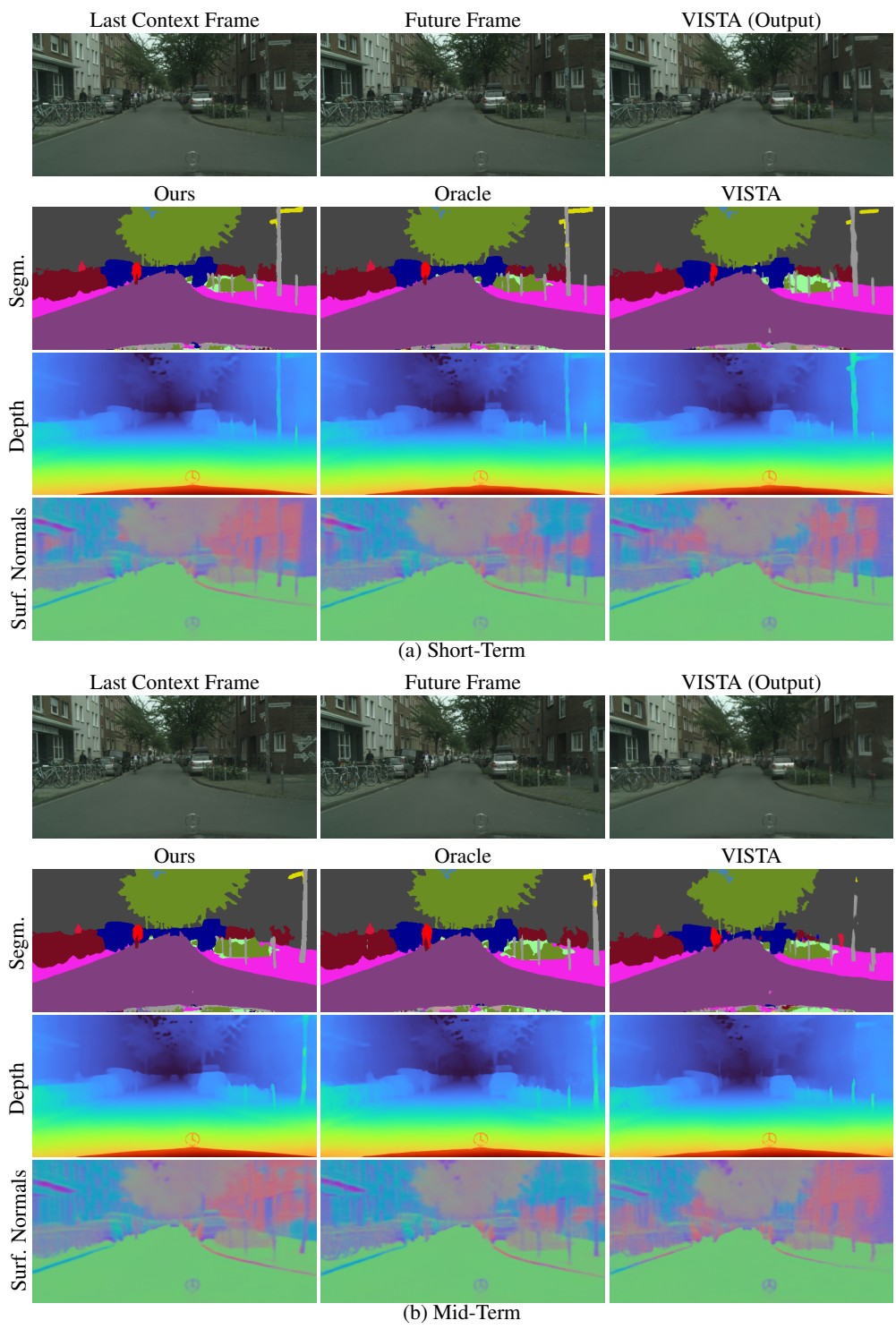

Figure 6: **Visualization of future predictions for semantic segmentation, depth, and surface normals**. The illustrated scene is Munster (15).

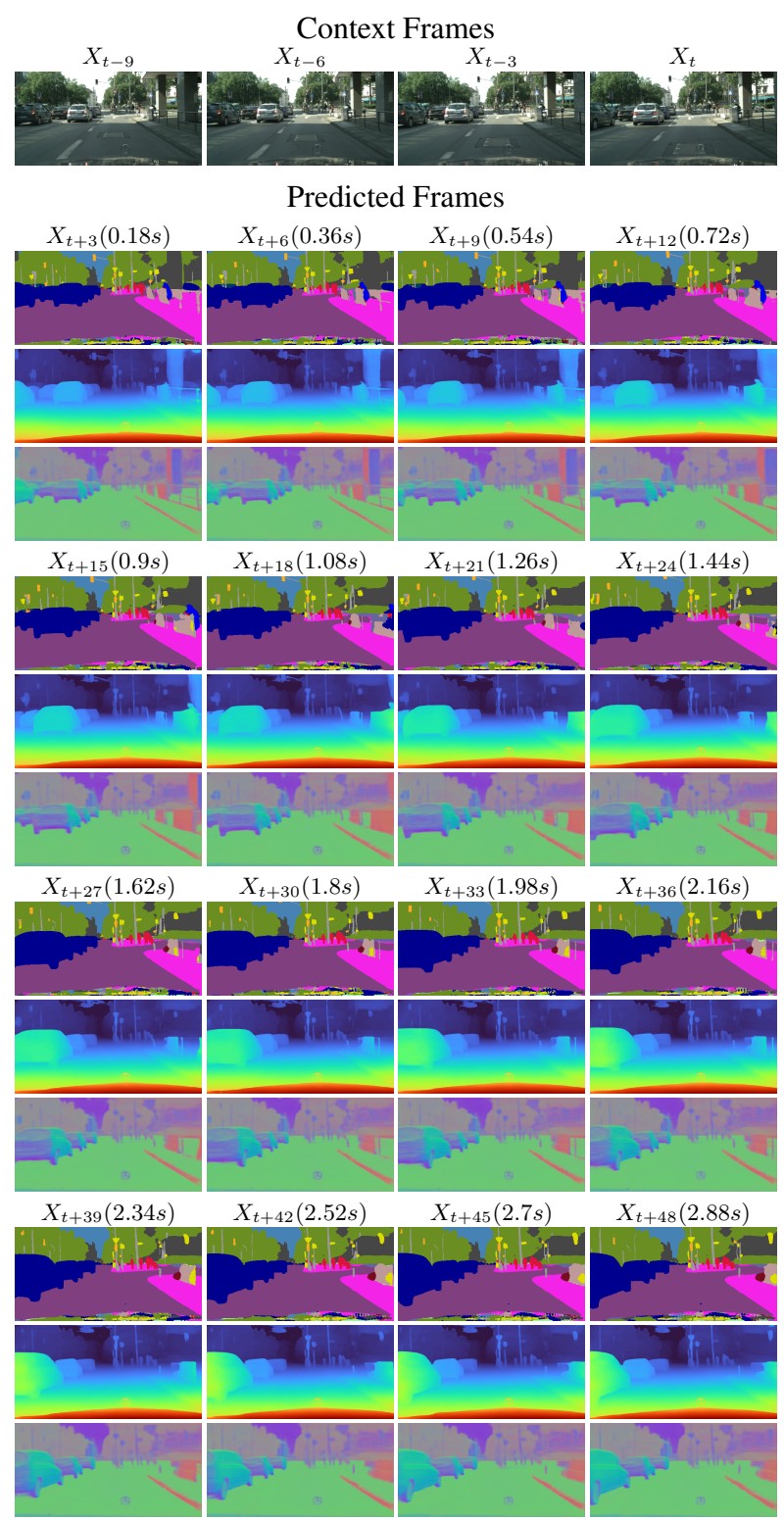

Figure 7: **Long-term semantic segmentation, depth and surface normal predictions.** The illustrated scene is Frankfurt (01 (011791-011820)). `DINO-Foresight` consistently preserves motion dynamics and intricate details in complex scenes over extended time horizons.

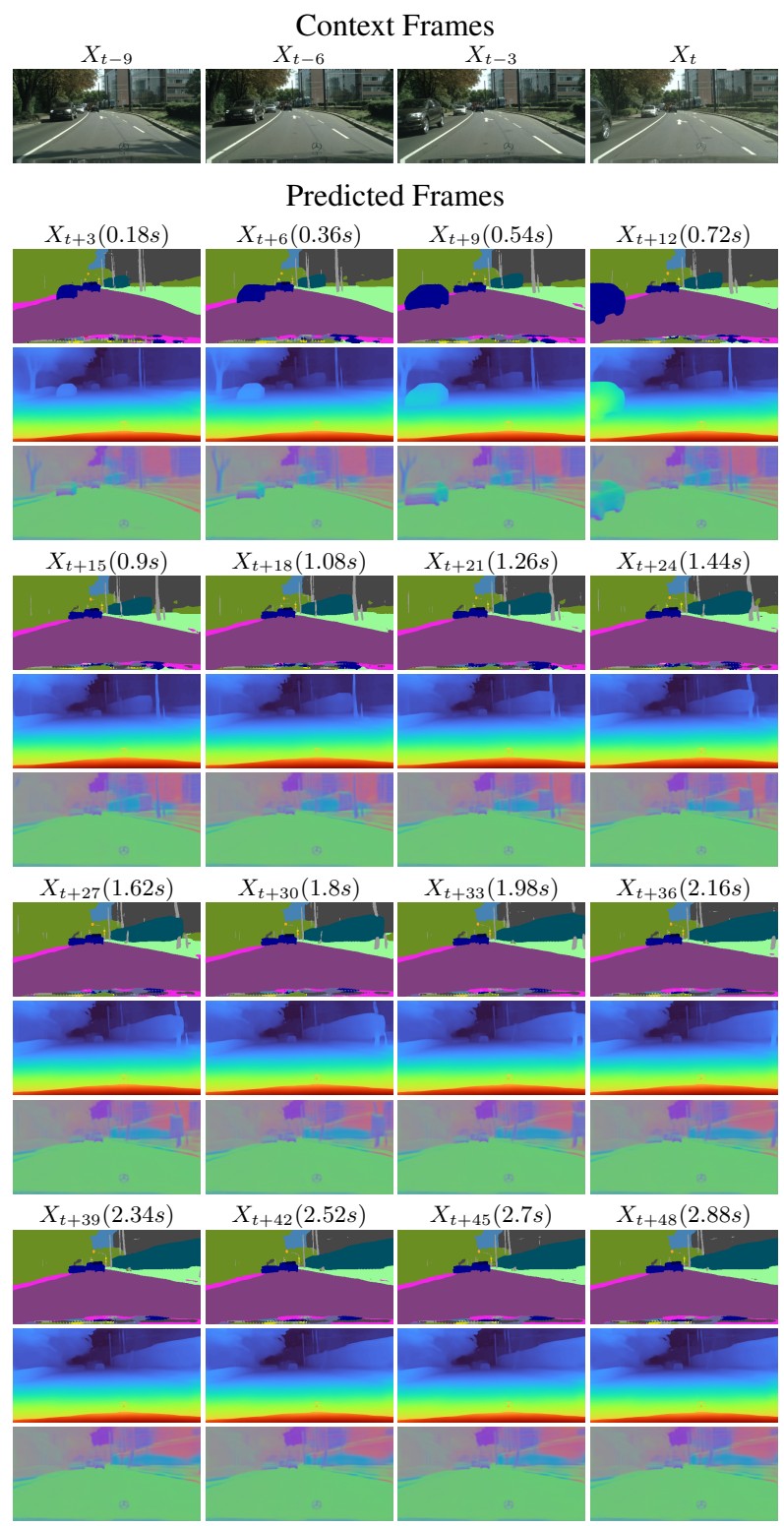

Figure 8: **Long-term semantic segmentation, depth and surface normal predictions**. The illustrated scene is Frankfurt (01 (006570-006599)). `DINO-Foresight` excels in predicting the motion of the nearby car but faces challenges with distant, low-motion objects, highlighting areas for future improvement.

