# OpenReview forum: "DINO-Foresight: Looking into the Future with DINO"
_NeurIPS.cc/2025/Conference — NeurIPS 2025 poster_

### Official Review · Reviewer_S83A · 2025-06-09

**Clarity:** 3
**Significance:** 3
**Originality:** 3
**Rating:** 5
**Confidence:** 4

**Summary:**

This paper introduces a novel framework for dynamic future prediction in autonomous driving. The authors propose operating in the feature space of pretrained Vision Foundation Models (VFMs) rather than at the pixel level like existing methods.
The approach casts dynamic future prediction as a self-supervised task with causal masking, where input tokens from previous frames are extracted using powerful VFMs and fed to a transformer. The model learns to predict latent tokens for future frames, which can then be decoded using off-the-shelf, task-specific prediction heads.
Results demonstrate competitive or superior performance compared to previous works while being significantly more efficient due to operating in compressed feature space rather than raw pixels.

**Questions:**

1) I wonder if a better approach would be to use a discrete pretrained tokenizer rather than a pretrained VFM, as this would make the training process much simpler: i.e., predict next tokens within a vocabulary size instead of continuous features after PCA. Why did the authors decide to use a continuous representation rather than a discrete one? What is the advantage?

2) Did the authors consider alterative masking strategies and different simple loss functions like simple L2/L1? It would be interesting to see how these affect the results.

**Ethical Concerns:**

["NO or VERY MINOR ethics concerns only"]

**Final Justification:**

After carefully reading the other reviews and the response to my concern, I've decided to raise my score to accept.
I strongly suggest the authors to better discuss the similarities with V-jepa however.

**Limitations:**

yes

**Paper Formatting Concerns:**

no formatting concerns

**Quality:**

3

**Strengths And Weaknesses:**

Strengths:
1. The paper is well-written and thoroughly ablated.
2. The addressed task is very interesting.
3. The idea of using only features from a single powerful VFM is smart and elegant, as it enables the possibility of quickly scaling to new modalities by simply training a new head on top of the original VFM features.

Weaknesses:
1. The authors claim that "VAE latents often lack semantic alignment, making them hard to interpret or use in downstream scene understanding." I think this is a strong claim, and in general, this is not true. There are many works that are actually successful in using VAE features. Although it is for images, works such "Massively Multimodal Masked Modeling" by Mizrahi et al., showed how VQ-VAE can provide useful features for solving many downstream tasks. I think it should be straightforward to apply discrete video tokenizers such as MAGVIT-v2 or MaskViT in the context of 4M to obtain a strong model that solves downstream tasks such as segmentation or depth on videos.
2. This work is very similar to V-JEPA, but viewed from the perspective of autonomous driving and downstream applications. Although I still believe this approach is interesting and looking at direct applications is useful, the paper should at least cite V-JEPA and highlight the main differences from an algorithmic perspective.
3. Futurist is cited as a competitor work and is listed in Table 1 with better results, but it is not discussed in Section 4.2.
4. Overall, I have the feeling that the paper is a composition of many different components from other papers, such as 2D-3D decomposition of attention, two-phase training, etc., and it lacks fundamental novel components.

---

> ### Author Rebuttal · Authors · 2025-07-30
>
> Thank you for reviewing our work and providing thoughtful and constructive feedback. Below, we provide our responses to each of your points:
>
> **Claims about VAEs**
>
> We appreciate the  point regarding VAE features and acknowledge the success of VAE-based approaches like 4M [2,3]. However, we highlight several important distinctions:
>
> First, our claim is that standard VAEs trained with image reconstruction objectives **achieve inferior semantic alignment in their latent spaces compared to powerful encoders like DINOv2**. To substantiate this claim, we conducted experiments using a publicly available VQ-VAE from the LDM framework [1]. As shown in Table 2 of our main paper, DPT heads trained on this VAE's latent embeddings significantly underperform compared to those using DINO and other VFMs in the oracle scenario. We further validated this by training a model to predict future discrete tokens within this VAE's vocabulary using cross-entropy loss, where predicted codes were converted to continuous features for downstream tasks. We will incorporate these findings in the relevant section (lines 322-326) of the main paper.
>
> Regarding 4M, we clarify that it employs separate VQ-VAE tokenizers for each modality, including one specifically trained for DINOv2 features (trained to reconstruct these features). This DINOv2 feature tokenizer differs fundamentally from standard image tokenizers. Our claim specifically concerns the limitations of image-based VAE tokenizers for scene understanding tasks. (As an aside, we note that our PCA reduction of DINOv2 features can be viewed as a simple continuous tokenizer alternative.)
>
> **Question 1: Discrete vs. Continuous features**
>
> In continuation to the previous discussion, and in response to the reviewer’s request, we compared forecasting performance using 4M's publicly available DINOv2 tokenizer (with cross-entropy loss on the predictor) against our proposed approach. For fair comparison, we retrained a simplified DINO-Foresight (ViT-B14 without registers, using only last-layer features). While the tokenized variant shows slightly worse oracle performance, DINO-Foresight without the VQ-VAE tokenization demonstrates significantly superior results for semantic future prediction.
>
> |  **Method**           |  **Short Term (mIoU↑)** |  **Short Term (MO_mIoU↑)** | **Mid Term (mIoU↑)** | **Mid Term (MO_mIoU↑)**  |
> |:--------------------------------|:----------------------:|:------------------------:|:--------------------:|:-----------------------:|
> | Oracle                          | 72.9                   | 74.0                     | 72.9                 | 74.0                    |
> | Oracle (4M tokenized)            | 70.2                   | 71.4                     | 70.2                 | 71.4                    |
> | Copy Last Baseline              | 54.7                   | 51.9                     | 40.5                 | 32.2                    |
> | Copy Last Baseline (4M tokenized)  | 53.7                   | 51.0                     | 40.0                 | 31.6                    |
> | Prediction                      | **68.9**                   | **69.3**                     | **57.3**                 | **55.0**                    |
> | Prediction (4M tokenized)          | 61.7                   | 60.9                     | 53.7                 | 51.0                    | 53.7                 | 51.0                    |
>
>
>
> **Comparison with V-JEPA**
>
> Thank you for highlighting this. We have cited V-JEPA in supplementary material and will be happy to add it to the main manuscript. However, there are key algorithmic differences: V-JEPA is a self-supervised pretraining method for learning general video representations (i.e., video encoders) via masked feature prediction, without any explicit focus on future feature forecasting (e.g., its evaluation is focused on video understanding tasks such as action classification). In contrast, our work specifically targets the future prediction of semantically rich features from frozen Vision Foundation Models (VFMs) to enable downstream future scene understanding. This reflects a fundamental difference in objectives: V-JEPA aims to learn transferable video representations, while our approach emphasizes multi-task semantic forecasting.
>
> Additionally, since V-JEPA's goal is to learn the video encoder, it jointly trains both encoder and predictor in an end-to-end manner, ultimately discarding the predictor after training. In contrast, Dino-Foresight keeps the VFM encoder frozen, training only the temporal predictor for forecasting. This design choice simplifies optimization and leverages the strong semantic consistency of DINOv2 features, making Dino-Foresight particularly well-suited for future prediction tasks, while V-JEPA remains more suitable for video understanding
>
> **Futurist discussion**
>
> You're right that we should better discuss Futurist. While Table 1 shows our competitive results, Futurist requires training separate models for each task (semantic segmentation and depth), whereas our single model handles all tasks simultaneously. This scalability advantage represents a key practical contribution (see provided discussion about advantages over futurist in the response to Reviewer Vuqi). We will expand this discussion in Section 4.2.
>
> **Technical novelty**
>
> While we build on established components, our core contribution is demonstrating that **forecasting VFM features** creates a unified semantic representation space for multi-task future prediction. This shifts the paradigm from task-specific or pixel-level forecasting to semantic feature forecasting. The 2D-3D attention decomposition and two-phase training are practical implementation choices rather than our primary novelties. Although these components are not novel themselves, we believe our work makes an important contribution by providing an accurate, open-source framework (code to be released) that will significantly advance research in feature-based future prediction.
>
>
> **Question 2: Alternative masking strategies and loss functions**
>
> We conducted extensive ablations on different loss functions and masking strategies, as shown in the tables below.
>
> Results demonstrate that different loss functions do not provide significant differences in performance:
>
> *Short Term Results:*
> | **Loss Function** | **Segm (mIoU↑)** | **Segm (MO_mIoU↑)** | **Depth (δ1↑)** | **Depth (AbsRel↓)** | **Normals (m↓)** | **Normals (11.25°↑)** |
> |:-----------------|:----------------:|:-------------------:|:----------------:|:--------------------:|:-------------------:|:------------------------:|
> | L1 | 71.7 | 71.7 | 88.6 | .118 | 3.41 | 94.3 |
> | MSE | 71.7 | **71.9** | 88.6 | .117 | 3.40 | **94.4** |
> | SmoothL1+cos | 71.7 | 71.4 | **88.7** | .116 | 3.40 | **94.4**|
> | SmoothL1 | **71.8** | 71.7 | 88.6 | **.114** | **3.39** | **94.4** |
>
> *Mid Term Results*
> | **Loss Function** | **Segm (mIoU↑)** | **Segm (MO_mIoU↑)** | **Depth (δ1↑)** | **Depth (AbsRel↓)** | **Normals (m↓)** | **Normals (11.25°↑)** |
> |:-----------------|:----------------:|:-------------------:|:----------------:|:--------------------:|:-------------------:|:------------------------:|
> | L1 | 59.7 | 57.6 | **85.6** | .138 | 4.00 | 91.3|
> | MSE | **60.0** | **57.8** | 85.4 | .138 | 3.99 | **91.4** |
> | SmoothL1+cos | 59.8 | 57.5 | 85.5 | .137 | **3.98** | **91.4** |
> | SmoothL1 | 59.8 | 57.6 | 85.4 | **.136** | 4.00 | 91.3 |
>
> For masking strategies, we implemented random masking for the future frame and compared it with our full masking approach. Our experiments demonstrate the superiority of the full masking strategy on future frame prediction over random masking.
>
> *Short Term Results*
> | **Masking Strategy**                | **Segm (mIoU↑)** | **Segm (MO_mIoU↑)** | **Depth (δ1↑)** | **Depth (AbsRel↓)** | **Normals (m↓)** | **Normals (11.25°↑)** |
> |:------------------------------|:---------------:|:------------------:|:---------------:|:-------------------:|:------------------------:|:-----------------------:|
> | Random Masking | 70.5            | 70.2               | 88.2            | .121                | 3.48                     | 94.0                    |
> | Full Masking     | **71.8**        | **71.7**           | **88.6**        | **.114**            | **3.39**                 | **94.4**                |
>
> *Mid Term Results*
> | **Masking Strategy**                | **Segm (mIoU↑)** | **Segm (MO_mIoU↑)** | **Depth (δ1↑)** | **Depth (AbsRel↓)** | **Normals (m↓)** | **Normals (11.25°↑)** |
> |:------------------------------|:---------------:|:------------------:|:---------------:|:-------------------:|:------------------------:|:-----------------------:|
> | Random Masking | 58.0            | 55.7               | 84.7            | .148                | 4.11                     | 90.8                    |
> | Full Masking     | **59.8**        | **57.6**           | **85.4**        | **.136**            | **4.00**                 | **91.3**                |
>
>
> [1]. High-Resolution Image Synthesis with Latent Diffusion Models
>
> [2]. 4M: Massively Multimodal Masked Modeling
>
> [3]. 4M-21: An Any-to-Any Vision Model for Tens of Tasks and Modalities

---

> > ### Comment · Reviewer_S83A · 2025-08-05
> > **Reply to author's rebuttal**
> >
> > I wish to thank the authors for their effort during this rebuttal and for thoroughly replying each of my concern.
> > I really appreciate the extensive analysis on the Continuous vs Discrete case. I believe this study is interesting and missing in other papers, and it should be included in the supplementary material.
> >
> > In conclusion, I've decided to raise my score to accept, as the provided answers are convincing to me.

---

### Official Review · Reviewer_Vuqi · 2025-06-30

**Clarity:** 2
**Significance:** 2
**Originality:** 2
**Rating:** 4
**Confidence:** 3

**Summary:**

The paper introduces DINO-Foresight, a self-supervised framework through forecasting the temporal evolution of VFM features (such as DINOv2), and it can be further adapted to various scene understanding tasks. Unlike prior video prediction methods that rely on generative models in pixel or VAE latent space, DINO-Foresight directly operates in the VFM feature space. The core idea is to predict the temporal evolution of rich, high-level VFM features using a masked transformer model, allowing seamless plug-and-play with downstream task-specific heads for applications like semantic segmentation, depth prediction, and instance segmentation. Besides, experimental results on several future prediction tasks demonstrate a unique advantage of DINO-Foresight.

**Questions:**

please see weaknesses

**Ethical Concerns:**

["NO or VERY MINOR ethics concerns only"]

**Final Justification:**

Thanks for the authors' detailed responses, which has essentially addressed my concerns. I am willing to raise my rating.

**Limitations:**

yes

**Quality:**

2

**Strengths And Weaknesses:**

**Strengths**

1. This paper leverages the strengths of VFMs and further improves them by forecasting the temporal evolution of VFM features, finally providing a powerful vision backbone.

2. The backbone can be seamlessly integrated with various task heads, and bring consistent improvements across these tasks.


**Weaknesses**
1. This paper builds upon existing models and designs, such as DINO and temporal MAE. Although the authors argue that DINO-Foresight is trained by directly forecasting VFM features, differing from standard MAE, it cannot adequately uphold the contribution's novelty.

2. The paragraph is not well-structured. For example, the paragraph in line 48 feels somewhat abrupt, and the paragraph in line 81 overemphasizes the differences.

3. Except for the feature prediction component, the model builds on prior designs, like FUTURIST, but the experiment results fail to demonstrate a significant improvement over these existing approaches.

---

> ### Author Rebuttal · Authors · 2025-07-30
>
> Thank you for reviewing our work and providing thoughtful and constructive feedback. Below, we provide our responses to each of your points:
>
> **Difference with DINOv2 and temporal MAE**
>
> While our method leverages DINOv2 features and employs masked transformer architectures, our core contribution centers on **modeling the temporal dynamics of VFM features to achieve accurate, multi-task, dense semantic future prediction**. In contrast, methods like Temporal MAE (or VideoMAE) focus on self-supervised pretraining of video representations (i.e., learning generic video encoders) through masked video modeling objectives. This represents a fundamental difference in goals: while their aim is to learn transferable video representations, ours is to enable precise future semantic forecasting.
>
> **Paragraph structure**
>
> We appreciate your feedback on the manuscript structure. Indeed, in both cases mentioned, we would like to emphasize our main contribution and highlight differences with latent generative approaches respectively. We will restructure the paragraphs to provide smoother transitions and clear flow.
>
> **Advantages over prior works**
>
> While Table 1 shows competitive numerical results with FUTURIST, there are important practical advantages that our method provides:
> 1) Unified framework: Our single model handles multiple tasks (semantic segmentation, depth, surface normals, instance segmentation) simultaneously using expressive and general frozen VFM representations, unlike prior works that would require task-specific training for each modality and dataset. For instance, FUTURIST would need to precompute pseudo-labels separately for each modality and dataset, significantly increasing implementation complexity.
> 2) Scalability: New tasks can be added by simply plugging in new heads that can be trained with limited labels, without fine-tuning the feature extractor or the prediction architecture. Furthermore, since Dino-Foresight operates directly on RGB inputs rather than precomputed modality-specific features, it can readily leverage additional raw video data for further improvements. This avoids potential misalignment issues that can arise when mixing precomputed features or pseudo-labels across datasets with differing label spaces or class definitions. Also, once FUTURIST is trained on specific modalities, it must be retrained from scratch to accommodate additional modalities.
> 3) DINO-Foresight is the only method to deliver future predictions for all four fundamental scene understanding tasks simultaneously, while being readily extensible to many more new tasks through its modular design. In contrast, prior works provide future predictions for to two tasks simultaneously at most.

---

> > ### Comment · Reviewer_Vuqi · 2025-08-05
> >
> > Thanks for the authors' detailed responses, which has essentially addressed my concerns. I am willing to raise my rating.

---

### Official Review · Reviewer_Bct7 · 2025-07-02

**Clarity:** 4
**Significance:** 3
**Originality:** 3
**Rating:** 5
**Confidence:** 4

**Summary:**

This paper introduces  DINO-Foresight, a self-supervised framework for semantic future prediction that leverages pretrained Vision Foundation Models (VFMs). The core idea of the framework is to forecast VFM features over time using a masked feature transformer. Unlike traditional approaches that rely on low-level pixel or latent space representations, DINO-Foresight directly predicts semantic features, which are then applied to various downstream tasks such as semantic segmentation, depth prediction, and surface normal estimation. The paper demonstrates that the proposed method outperforms state-of-the-art methods in future-frame prediction across various tasks.

**Questions:**

Please see the Weaknesses above.

**Ethical Concerns:**

["NO or VERY MINOR ethics concerns only"]

**Final Justification:**

Accept: Technically solid paper with a good rebuttal that addresses my previous concerns.

**Limitations:**

yes

**Paper Formatting Concerns:**

No major formatting issue.

**Quality:**

3

**Strengths And Weaknesses:**

Strengths:
- This paper is well-motivated, and the writing is easy to follow.
- Extensive experiments on multi-task performance. In Table1, the authors compare their method with multiple counterparts on several forecasting tasks, including depth estimation and instance segmentation. Such comprehensive comparisons greatly demonstrate the superiority of the proposed method.
- The idea of predicting semantic features (from vision foundation models) of future frames enables more efficient learning and inference by omitting the generation of low-level pixel details. This solution might be practically useful and insightful for research areas that are sensitive to computational efficiency, such as real-time autonomous driving.

Weaknesses:
- Lack of discussion with some prior works. Besides the related work listed in Semantic Future Prediction (in lines 86-99), some prior works are missing from the discussion and comparison, such as V-JEPA and DINO-WM.

- Lack of investigation on the scalability of the model size and data scale. Experiments are conducted on a fixed model size and data scale without any indication of how task performance would be with more/less parameters and data.


References:
V-JEPA: Revisiting Feature Prediction for Learning Visual Representations from Video

DINO-WM: DINO-WM: World Models on Pre-trained Visual Features enable Zero-shot Planning

---

> ### Author Rebuttal · Authors · 2025-07-30
>
> Thank you for reviewing our work and providing thoughtful and constructive feedback. Below, we provide our responses to each of your points:
>
> **Discussion with some prior works**
>
> Thank you for raising this important point. We acknowledge our manuscript could benefit from a more thorough discussion of additional related methods such as V-JEPA and DINO-WM.
>
> - V-JEPA is a self-supervised pretraining method for learning general video representations (i.e., video encoders) via masked feature prediction, without explicit focus on future feature forecasting. In contrast, our work specifically targets the prediction of semantically rich features from frozen Vision Foundation Models (VFMs) to enable downstream future scene understanding. This reflects a fundamental difference in objectives: V-JEPA aims to learn transferable video representations, while our approach emphasizes multi-task semantic forecasting.
> - DINO-WM utilizes DINO features for world modeling, primarily applied to action-conditioned planning in simulated environments with synthetic data. Our work, instead, focuses on real-world autonomous driving scenarios and multi-task future prediction, with an emphasis on accurate dense (pixel-wise) semantic forecasting rather than planning. We would like also to mention that these two works were developed concurrently.
>
> We will revise the manuscript to incorporate these discussions.
>
> **Model size and Data Scale scalability**
>
> You raise an excellent point about scalability analysis. While our initial experiments focus on demonstrating the core concept with a fixed architecture, investigating how performance scales with model size and data would provide valuable insights into our method's potential. To this end, we provide preliminary results of model size and data scale experiments conducted to assess the impact on downstream task performance:
>
> The following two tables demonstrate how varying the model size—while keeping dataset size and training duration fixed—affects downstream task performance for both short-term and mid-term future prediction. We evaluate a small and a large variant alongside our default (base) architecture by modifying the hidden dimension and number of attention heads. Results indicate that our framework benefits from increased model capacity, with improvements particularly notable for mid-term predictions. These findings motivate future work to explore even larger variants and leverage more data to fully realize the potential of forecasting VFM features for multi-task scene understanding.
>
> *Short Term Results:*
>
> | Model Variant      | Hidden Dim | Att Heads | Segm (mIoU↑) | Segm (MO_mIoU↑) | Depth (δ1↑) | Depth (AbsRel↓) | Normals (m↓) | Normals (11.25°↑) |
> |:------------------|:----------:|:---------:|:------------:|:---------------:|:-----------:|:--------------:|:------------:|:----------------:|
> | Oracle             | -          | -         | 77.0         | 77.4            | 89.1        | 0.108          | 3.24         | 95.3             |
> | Copy Last Baseline | -          | -         | 54.7         | 52.0            | 84.1        | 0.154          | 4.41         | 89.2             |
> | Small (115 M)      | 768        | 6         | 71.1         | 70.8            | 87.7        | 0.125          | 3.58         | 93.7             |
> | Base (258 M)       | 1152       | 8         | 71.8         | **71.7**            | **88.6**        | **0.114**          | 3.39         | **94.4**             |
> | Large (460 M)      | 1536       | 12        | **71.9**         | **71.7**            | **88.6**        | 0.116          | **3.40**        | **94.4**             |
>
> *Mid Term Results:*
> | Model Variant      | Hidden Dim | Att Heads | Segm (mIoU↑) | Segm (MO_mIoU↑) | Depth (δ1↑) | Depth (AbsRel↓) | Normals (m↓) | Normals (11.25°↑) |
> |:------------------|:----------:|:---------:|:------------:|:---------------:|:-----------:|:--------------:|:------------:|:----------------:|
> | Oracle             | -          | -         | 77.0         | 77.4            | 89.1        | 0.108          | 3.24         | 95.3             |
> | Copy Last Baseline | -          | -         | 40.4         | 32.3            | 77.8        | 0.212          | 5.39         | 84.0             |
> | Small (115 M)      | 768        | 6         | 59.3         | 57.3            | 84.8        | 0.142          | 4.13         | 90.8             |
> | Base (258 M)       | 1152       | 8         | 59.8         | 57.6            | 85.4        | **0.136**          | **4.00**         | 91.3             |
> | Large (460 M)      | 1536       | 12        | **60.2**         | **58.3**           | **85.5**        | 0.137          | **4.00**         | **91.5**            |
>
>
> The following two tables illustrate the impact of varying training data size—while keeping model size fixed—on downstream task performance for both short-term and mid-term future prediction. Specifically, we combine the Cityscapes (CS) and nuScenes (NU) datasets, ensuring equal-probability sampling from each during training. Preliminary results on the Cityscapes test set show that the model trained on the combined datasets achieves consistent improvements in semantic future prediction across all tasks. These findings highlight the benefit of increased and diverse training data and motivate further exploration of scalability with larger datasets and extended training to better realize the potential of forecasting VFM features for multi-task scene understanding.
>
> *Short Term Results:*
>
> | Training data               | Segm (mIoU↑) | Segm (MO_mIoU↑) | Depth (δ1×100↑) | Depth (AbsRel↓) | Normals (Mean Ang Err↓) | Normals (11.25°↑) |
> |:----------------------------|:-------------:|:---------------:|:---------------:|:---------------:|:-----------------------:|:-----------------:|
> | Oracle           | 77.0         | 77.4            | 89.1        | 0.108          | 3.24         | 95.3 |
> | Copy Last Baseline | 54.7         | 52.0            | 84.1        | 0.154          | 4.41         | 89.2             |
> | Cityscapes    | 71.8          | 71.7            | **88.6**          | **0.114**           | 3.39                    | **94.4**             |
> | Cityscapes+Nuscenes | **72.3**          | **72.2**            | **88.6**           | 0.117           | 3.41                    | **94.4**           |
>
> *Mid Term Results:*
> | Training data                | Segm (mIoU↑) | Segm (MO_mIoU↑) | Depth (δ1×100↑) | Depth (AbsRel↓) | Normals (Mean Ang Err↓) | Normals (11.25°↑) |
> |:----------------------------|:-------------:|:---------------:|:---------------:|:---------------:|:-----------------------:|:-----------------:|
> | Oracle           | 77.0         | 77.4            | 89.1        | 0.108          | 3.24         | 95.3    |
> | Copy Last Baseline |  40.4         | 32.3            | 77.8        | 0.212          | 5.39         | 84.0             |
> | Cityscapes    | 59.8          | 57.6            | 85.4            | **0.136**           | 4.00                    | 91.3             |
> | Cityscapes+Nuscenes | **61.0**          | **59.4**            | **85.7**            | **0.136**          | **3.95**                    | **91.6**            |

---

> > ### Comment · Reviewer_Bct7 · 2025-08-06
> >
> > Thanks for the detailed response. The added discussion with prior works and more experimental results on model scalability have sufficiently addressed my previous concerns. Therefore, I'm willing to raise my rating. Please include related discussions and results in the revision.

---

### Official Review · Reviewer_BMEg · 2025-07-02

**Clarity:** 3
**Significance:** 3
**Originality:** 3
**Rating:** 6
**Confidence:** 4

**Summary:**

DINO-Foresight introduces a self-supervised framework for predicting future scene dynamics by forecasting vision foundation model (DINOv2) features of future frames of videos. A key contribution is the use of high resolution features from multiple layers of the DINO model and the use of PCA for dimensionality reduction. This allows the model to predict future depth maps, semantic segmentation maps and surface normals of future frames of videos from steet scene datasets: CityScapes and nuScenes.

**Questions:**

•	The paper should highlight its novel technical contributions in more detail.
•	Does the proposed approach generalize to novel scenarios?
•	Can the proposed approach handle scenarios where the distribution of future states is multi-modal?

**Ethical Concerns:**

["NO or VERY MINOR ethics concerns only"]

**Final Justification:**

The rebuttal addressed all my concerns.

**Limitations:**

Yes.

**Paper Formatting Concerns:**

None.

**Quality:**

3

**Strengths And Weaknesses:**

**Strengths:**

•	The paper demonstrates state of the art results on a wide range of tasks ranging from the prediction of future semantic segmentation maps to depth maps.

•	The proposed method is simple and computationally efficient.

•	The paper includes adequate ablations highlighting the effectiveness of the proposed approach.

•	The paper is well written and easy to understand.

**Weakness:**

•	Novelty: The proposed method is relatively simple – it predicts the future in the feature space instead of RGB/semantic/depth pixel space. The key contribution is the integration of high level features using a PCA dimensionality reduction step. A key downside of predictions in the DINO feature space is the inability to predict future RGB frames as DINO features still discard a lot of high-frequency details in the original RGB input sequence.

•	Generalization: The proposed model is trained and test on the same datasets: Cityscapes and nuScenes. However, the source DINO model is inherently generalizable across datasets. It would be interesting to see zero-shot applications to datasets such as Waymo Open in the street scene domain.

•	Evaluation metrics: In case of street scenes, nearby objects have a larger apparent motion, while faraway objects can appear static. The paper uses evaluation metrics that do not take such motion into account. While this is also an issue with prior work, it would be interesting to use a motion aware evaluation metric, to highlight that the proposed model can predict motion correctly.

•	 Multi-modality: The distribution of future, e.g., in case of street scenes, is highly multi-modal – there are many distinct likely futures, i.e., a vehicle can turn to the left or to the right. The proposed model cannot capture this multi-modal distribution.

•	Diverse scenarios: The proposed model is limited to street scenes, it is not clear if the model can be applied to arbitrary videos for future frame prediction. It would be interesting to include evaluation on diverse domains.

---

> ### Author Rebuttal · Authors · 2025-07-31
>
> Thank you for your comprehensive review and insightful feedback. We appreciate the assessment of our work and the constructive suggestions for improvement. Below, we provide our responses to each of your points:
>
> **Clarifying the technical contribution of our work**
>
> We appreciate your assessment of the method's simplicity as a strength while noting concerns about novelty. Our core contribution lies in demonstrating that modeling the temporal evolution of VFM features enables effective multi-task semantic future prediction. While the individual technical components are not novel, our design reveals that VFM features maintain semantic richness when predicted temporally, enabling a unified framework that operates across multiple scene understanding tasks without task-specific training.
>
> **Predicting future RGB frames from DINO**
>
> While out of scope for this work, we explored the ability of our method to predict future RGB from predicted DINO features. To this end, we trained offline a DPT head with simple L2 loss to "reconstruct" RGB from DINOv2 features using just ~3000 RGB images and then applied it to our prediction transformer following our pipeline. Preliminary results show that in the oracle setting, DINOv2+DPT can sufficiently reconstruct RGB frames with very low MSE, and in the prediction setting provides sufficient RGB predictions with clear structure but blurry regions in high motion/ambiguous areas. This preliminary experiment demonstrates that while our focus remains on semantic prediction for autonomous driving applications, our framework shows promising potential for extension to RGB prediction - potentially through a generative decoder (e.g., diffusion models) that could convert predicted DINO features to RGB frames.
>
> **Zero-shot Evaluation**
>
> You raise a valuable point regarding the importance of cross-dataset evaluation. While our primary analysis has focused on Cityscapes and nuScenes, the model is built on top of DINOv2 features, which are inherently designed for strong generalizability across diverse visual domains. Due to time constraints and the lack of standardized benchmarks and data loading modules for Waymo (Panoramic Video Panoptic Segmentation Subset), we provide preliminary zero-shot results on nuScenes validation set using a model trained solely on Cityscapes. Notably, the zero-shot performance on nuScenes is only slightly worse than that of a model trained directly on nuScenes and surpasses all baselines. These initial findings demonstrate encouraging potential for zero-shot transfer beyond the original training dataset, supporting the view that our approach can generalize effectively to new domains. We see comprehensive cross-dataset and zero-shot evaluations as an important direction for future work.
>
> *Short Term Results on Nuscenes*
>
> | Method              | Depth (δ1↑) | Depth (AbsRel↓) | Normals (m↓) | Normals (11.25°↑) |
> | :---               | :---:       | :---:           | :---:         | :---:              |
> | Oracle              | 82.6        | 0.206           | 3.09          | 97.1               |
> | Copy Last Baseline  | 73.4        | 0.353           | 4.66          | 88.6               |
> | Vista               | 74.6        | 0.337           | 4.46          | 90.8               |
> | Nuscenes Trained    | 80.7        | 0.218           | 3.59          | 93.9               |
> | Cityscapes Trained  | 78.4        | 0.269           | 4.03          | 92.3               |
>
>
> *Mid Term Results on Nuscenes*
>
> | Method              | Depth (δ1↑) | Depth (AbsRel↓) | Normals (m↓) | Normals (11.25°↑) |
> | :---               | :---:       | :---:           | :---:         | :---:              |
> | Oracle              | 82.6        | 0.206           | 3.09          | 97.1               |
> | Copy Last Baseline  | 69.5        | 0.442           | 5.15          | 86.3               |
> | Vista               | 70.8        | 0.421           | 4.96          | 88.2               |
> | Nuscenes Trained    | 76.0        | 0.299           | 4.27          | 90.5               |
> | Cityscapes Trained  | 73.7        | 0.346           | 4.54          | 89.8               |
>
> **Motion-aware evaluation**
>
> We appreciate the suggestion to incorporate motion-aware evaluation metrics. In this work, we have followed standard benchmarks and evaluation protocols established by prior works, including MO-mIoU, which measures segmentation performance specifically on movable objects and thus partially addresses the dynamic aspects of street scenes. Incorporating motion-weighted evaluation techniques would be an interesting addition to assess the model’s ability to predict dynamic scene elements. To the best of our knowledge, such a comparison or implementation is not available in the standard benchmarks we adopted. However, due to the limited available time for the rebuttal, we were unable to implement such metric at this point. We are open to adopting such metrics in the final version of the paper or as part of future work.
>
> **Multi-modality**
>
> You correctly identify that our current approach produces deterministic predictions rather than capturing the multi-modal nature of future possibilities. We acknowledge this limitation in our paper's limitations section, where we discuss that our deterministic architecture does not account for inherent uncertainty in future predictions and propose incorporating stochastic elements such as diffusion losses on tokens to better capture this ambiguity. Despite predicting the most plausible future based on observed conditions, it does not capture the full distribution of possible outcomes. This represents an important future direction for handling scenarios where multiple futures are equally plausible (e.g., vehicles at intersections).
>
> **Domain diversity**
>
> We acknowledge that our evaluation is currently limited to street scenes, which constrains broader claims about general applicability. However, our approach operates on frozen DINOv2 VFM features, which are inherently generalizable and not domain-specific, having been trained on diverse visual data from multiple domains. By conducting temporal modeling directly in the VFM feature space, we further reduce the domain gap that would typically exist in raw RGB space, increasing the likelihood that our method can transfer to varied input domains beyond autonomous driving. Thus, extending our evaluation to more diverse video datasets is an important direction for future work. .

---

> ### Comment · Reviewer_BMEg · 2025-08-04
> **Great Paper!**
>
> Overall the rebuttal has addressed all my concerns. In particular, It is great to see the zero shot generalization results! I would strongly vote for acceptance.
>
> For the final version, I would encourage the authors to include a discussion on V-JEPA.

---

### Decision · Program_Chairs · 2025-09-17

**Decision:**

Accept (poster)

**Comment:**

This submission received all four positive scores from the reviewers (1xBA, 2xA, 1xSA). The reviewers especially appreciated simplicity and effectiveness of the approach, together with thorough evaluations. The remaining questions were addressed in the rebuttal. The final recommendation is therefore to accept.